# Nearly Minimax Optimal Offline Reinforcement Learning with Linear Function Approximation: Single-Agent MDP and Markov Game

**Wei Xiong**[*1]**, Han Zhong**[*2]**, Chengshuai Shi**[3]**, Cong Shen**[3]**, Liwei Wang**[4,5]**, Tong Zhang**[1,6]

Department of Mathematics, The Hong Kong University of Science and Technology[1]
Center for Data Science, Peking University[2]
Department of Electrical and Computer Engineering, University of Virginia[3]
National Key Laboratory of General Artificial Intelligence, Peking University[4]
School of Intelligence Science and Technology, Peking University[5]
Department of Computer Science and Engineering, The Hong Kong University of Science and Technology[6]
{wxiongae, tongzhang}@ust.hk; hanzhong@stu.pku.edu.cn;
{cs7ync, cong}@virginia.edu; wanglw@cis.pku.edu.cn

## Abstract

Offline reinforcement learning (RL) aims at learning an optimal strategy using a pre-collected dataset without further interactions with the environment. While various algorithms have been proposed for offline RL in the previous literature, the minimax optimality has only been (nearly) established for tabular Markov decision processes (MDPs). In this paper, we focus on offline RL with linear function approximation and propose a new pessimism-based algorithm for offline linear MDP. At the core of our algorithm is the uncertainty decomposition via a reference function, which is new in the literature of offline RL under linear function approximation. Theoretical analysis demonstrates that our algorithm can match the performance lower bound up to logarithmic factors. We also extend our techniques to the two-player zero-sum Markov games (MGs), and establish a new performance lower bound for MGs, which tightens the existing result, and verifies the nearly minimax optimality of the proposed algorithm. To the best of our knowledge, these are the first computationally efficient and nearly minimax optimal algorithms for offline single-agent MDPs and MGs with linear function approximation.

## 1 Introduction

Reinforcement learning (RL) has achieved tremendous empirical success in both single-agent (Kober et al., 2013) and multi-agent scenarios (Silver et al., 2016; 2017). Two components play a critical role – function approximations and efficient simulators. For RL problems with a large (or even infinite) number of states, storing a table as in classical Q-learning is generally infeasible. In these cases, practical algorithms (Mnih et al., 2015; Lillicrap et al., 2015; Schulman et al., 2015; 2017; Haarnoja et al., 2018) approximate the true value function or policy by a function class (e.g., neural networks). Meanwhile, an efficient simulator allows for learning a policy in an online trial-and-error fashion using millions or even billions of trajectories. However, due to the limited availability of data samples in many practical applications, e.g., healthcare (Wang et al., 2018) and autonomous driving (Pan et al., 2017), instead of collecting new trajectories, we may have to extrapolate knowledge only from past experiences, i.e., a pre-collected dataset. This type of RL problems is usually referred to as *offline RL* or *batch RL* (Lange et al., 2012; Levine et al., 2020).

An offline RL algorithm is usually measured by its sample complexity to achieve the desired statistical accuracy. A line of works (Xie et al., 2021b; Shi et al., 2022; Li et al., 2022) demonstrates that near-optimal sample complexity in *tabular* single-agent MDPs is attainable. However, these algorithms cannot solve the problem with large or infinite state spaces where function approximation is involved. To our best knowledge, existing algorithms cannot attain the statistical limit even for *linear function*

---

*The first two authors contribute equally.

*approximation*, which is arguably the simplest function approximation setting. Specifically, for linear function approximation, Jin et al. (2021c) proposes the first efficient algorithm for offline linear MDPs, but their upper bound is suboptimal compared with the existing lower bounds in Jin et al. (2021c); Zanette et al. (2021). Recently, Yin et al. (2022) tries to improve the result by incorporating variance information in the algorithmic design of offline MDPs with linear function approximation. However, a careful examination reveals a technical gap, and some additional assumptions may be needed to fix it (cf. Section 3 and Section 5). Beyond the single-agent MDPs, Zhong et al. (2022) studies the Markov games (MGs) with linear function approximation and provides the only provably efficient algorithm with a suboptimal result. Therefore, the following problem remains open:

> *Can we design computationally efficient offline RL algorithms for problems with linear function approximation that are nearly minimax optimal?*

In this paper, we first answer this question affirmatively under linear MDPs (Jin et al., 2020) and then extend our results to the two-player zero-sum Markov games (MGs) (Xie et al., 2020). Our contributions are summarized as follows:

- We identify an implicit and restrictive assumption required by existing approaches in the literature, which originates from omitting the complicated temporal dependency between different time steps. See Section 3 for a detailed explanation.

- We handle the temporal dependency by an uncertainty decomposition technique via a reference function, thus closing the gap to the information-theoretic lower bound without the restrictive independence assumption. The uncertainty decomposition serves to avoid a $\sqrt{d}$-amplification of the value function error and also the measurability issue from incorporating variance information to improve the $H$-dependence, where $d$ and $H$ are the feature dimension and planning horizon, respectively. To the best of our knowledge, this technique is new in the literature of offline learning under linear function approximation.

- We further generalize the developed techniques to two-player zero-sum linear MGs (Xie et al., 2020), thus demonstrating the broad adaptability of our methods. Meanwhile, we establish a new performance lower bound for MGs, which tightens the existing results, and verifies the nearly minimax optimality of the proposed algorithm.

## 1.1 RELATED WORK

Due to space limit, we defer a comprehensive review of related work to Appendix A.2 but focus on the works that are most related to the problem setup and our algorithmic designs.

**Offline RL with Linear Function Approximation**. Jin et al. (2021c) and Zhong et al. (2022) provide the first results for offline linear MDPs and two-player zero-sum linear MGs, respectively. However, their algorithms are based on Least-Squares Value Iteration (LSVI), and establish pessimism by adding bonuses at every time step thus suffering from a $\sqrt{d}$-amplification to the lower bound (Zanette et al., 2021; Zhong et al., 2022). The amplification results from the statistical dependency between different time steps. After these, Min et al. (2021) studies the offline policy evaluation (OPE) in linear MDP with an additional independence assumption that the data samples between different time steps are independent thus circumventing this core issue. Yin et al. (2022) studies the policy optimization in linear MDP, which also (implicitly) require the independence assumption. Another line of work addresses the error amplification from temporal dependency with different algorithmic designs. Zanette et al. (2020) designs an actor-critic-based algorithm and establishes pessimism via direct perturbations of the parameter vectors in a linear function approximation scheme. Xie et al. (2021a); Uehara and Sun (2021) establish pessimism only at the *initial* state but at the expense of computational tractability. These algorithmic ideas are fundamentally different from ours and do not apply to the LSVI-type algorithms. We will compare the proposed algorithms with them in Section 5.

## 2 OFFLINE LINEAR MARKOV DECISION PROCESS

**Notations**. Given a semi-definite matrix $\Lambda$ and a vector $u$, we denote $\sqrt{u^\top \Lambda u}$ as $\|u\|_\Lambda$. The 2-norm of a vector $w$ is $\|w\|_2$. We also denote $\lambda_{\min}(A)$ as the smallest eigenvalue of the matrix $A$. The subscript of $f(x)_{[0,M]}$ means that we clip the value $f(x)$ to the range of $[0, M]$, i.e.,

$f(x)_{[0,M]} = \max\{0, \min\{M, f(x)\}\}$. Given a set $X$, we define the set of probability measure on it by $\Delta_X$. We will use the shorthand notations $\phi_h = \phi(x_h, a_h)$, $\phi_h^\tau = \phi(x_h^\tau, a_h^\tau)$, $r_h = r_h(x_h, a_h)$, and $r_h^\tau = r_h(x_h^\tau, a_h^\tau)$ (formally defined in the next subsection). With a slight abuse of notations, we also use similar notations (e.g. $\phi_h = \phi(x_h, a_h, b_h)$) for MGs, which shall be clear from the context. $Y \lesssim X$ means $Y \leq CX$ for some constant $C > 0$. To improve readability, we also provide a summary of notations in Appendix A.

**Markov Decision Process**. We consider an episodic MDP, denoted as $\mathcal{M}(\mathcal{S}, \mathcal{A}, H, \mathbb{P}, r)$, where $\mathcal{S}$ and $\mathcal{A}$ are the state space and action space, $H$ is the episode length, $\mathbb{P} = \{\mathbb{P}_h\}_{h=1}^H$ and $r = \{r_h\}_{h=1}^H$ are the state transition kernels and reward functions, respectively. For each $h \in [H]$, $\mathbb{P}_h(\cdot|x, a)$ is the distribution of the next state given the state-action pair $(x, a)$ at step $h$, $r_h(x, a) \in [0, 1]$ is the deterministic reward given the state-action pair $(x, a)$ at step $h$. [1]

**Policy and Value function.** A policy $\pi = \{\pi_h\}_{h=1}^H$ is a collection of mappings from a state $x \in \mathcal{S}$ to a distribution of action space $\pi_h(\cdot|x) \in \Delta_\mathcal{A}$. For any policy $\pi$, we define the Q-value function $Q_h^\pi(x, a) = \mathbb{E}_\pi[\sum_{h'=h}^H r_{h'}(x_{h'}, a_{h'})|(x_h, a_h) = (x, a)]$ and the V-value function $V_h^\pi(x) = \mathbb{E}_\pi[\sum_{h'=h}^H r_{h'}(x_{h'}, a_{h'})|x_h = x]$, where $a_{h'} \sim \pi_{h'}(\cdot|x_{h'})$ and $x_{h'+1} \sim \mathbb{P}_{h'}(\cdot|x_{h'}, a_{h'})$. For any function $V : \mathcal{S} \to \mathbb{R}$, we denote the conditional mean as $(\mathbb{P}_h V)(x, a) := \sum_{x' \in \mathcal{S}} \mathbb{P}_h(x'|x, a)V(x')$ and conditional variance as $[\text{Var}_h V](x, a) := [\mathbb{P}_h V^2](x, a) - ([\mathbb{P}_h V](x, a))^2$. The Bellman operator is defined as $(\mathcal{T}_h V)(x, a) := r_h(x, a) + (\mathbb{P}_h V)(x, a)$.

We consider the MDPs whose rewards and transitions possess a linear structure (Jin et al., 2020).

**Definition 1** (Linear MDP). *MDP$(\mathcal{S}, \mathcal{A}, H, \mathbb{P}, r)$ is a linear MDP with a (known) feature map $\phi : \mathcal{S} \times \mathcal{A} \to \mathbb{R}^d$, if for any $h \in [H]$, there exist $d$ unknown signed measures $\mu_h = (\mu_h^{(1)}, \cdots, \mu_h^{(d)})$ over $\mathcal{S}$ and an unknown vector $\theta_h \in \mathbb{R}^d$, such that for any $(x, a) \in \mathcal{S} \times \mathcal{A}$, we have $\mathbb{P}_h(\cdot \mid x, a) = \langle \phi(x, a), \mu_h(\cdot) \rangle$, $r_h(x, a) = \langle \phi(x, a), \theta_h \rangle$. With loss of generality, we assume that $\|\phi(x, a)\| \leq 1$ for all $(x, a) \in \mathcal{S} \times \mathcal{A}$, and $\max\{\|\mu_h(\mathcal{S})\|, \|\theta_h\|\} \leq \sqrt{d}$ for all $h \in [H]$.*

For linear MDP, we have the following result, whose proof can be found in Jin et al. (2020).

**Lemma 1.** *For any function $V : \mathcal{S} \to [0, V_{\max} - 1]$ and $h \in [H]$, there exist vectors $\beta_h, w_h \in \mathbb{R}^d$ with $\|\beta_h\| \leq \|w_h\| \leq \sqrt{d}V_{\max}$, such that $\forall(x, a) \in \mathcal{S} \times \mathcal{A}$ such that the conditional expectation and Bellman equation are both linear in the feature:*

$$(\mathbb{P}_h V)(x, a) = \phi(x, a)^\top \beta_h, \quad and \quad (\mathcal{T}_h V)(x, a) = \phi(x, a)^\top w_h. \tag{1}$$

**Offline RL.** In offline RL, the algorithm needs to learn a near-optimal policy or to approximate Nash Equilibrium (NE) from a pre-collected dataset without further interacting with the environment. We suppose that we have access to a batch dataset $\mathcal{D} = \{x_h^\tau, a_h^\tau, r_h^\tau : h \in [H], \tau \in [K]\}$, where each trajectory is independently sampled by a behavior policy $\mu$. The induced distribution of the state-action pair at step $h$ is denoted as $d_h^b$. We make the following standard dataset coverage assumption for offline RL with linear function approximation (Wang et al., 2020; Duan et al., 2020):

**Assumption 1.** *We assume $\kappa = \min_{h \in [H]} \lambda_{\min}(\mathbb{E}_{d_h^b}[\phi(x, a)\phi(x, a)^\top]) > 0$ for MDPs.*

This assumption requires the behavior policy to explore the state-action space well and is not information-theoretic as Jin et al. (2021c) does not necessarily require it. However, we make this assumption so that we can employ the variance information, which seems challenging otherwise. We remark the assumption is also made by the existing works that employ the variance information for the offline RL problems with linear function approximation (Min et al., 2021; Yin et al., 2022).

## 3 PRELIMINARIES AND TECHNICAL CHALLENGES

**Pessimistic Value Iteration (PEVI)** is proposed in the seminal work of Jin et al. (2021c), which constructs the value function estimations $\{\widehat{V}_h(\cdot)\}_{h=1}^H$ and Q-value estimations $\{\widehat{Q}_h(\cdot, \cdot)\}_{h=1}^H$ backward from $h = H$ to $h = 1$, with the initialization $\widehat{V}_{H+1} = 0$. Specifically, given $\widehat{V}_{h+1}$, PEVI

---

[1]The results readily generalize to the stochastic reward as the uncertainty of reward is non-dominating compared with that of state transition.

approximates the Bellman equation by ridge regression:

$$\widehat{Q}_h(x,a) \leftarrow \widehat{\mathcal{T}}_h \widehat{V}_{h+1}(x,a) - \beta \left\| \phi(x,a) \right\|_{\Lambda_h^{-1}},$$

where the linear approximation $\widehat{\mathcal{T}}_h \widehat{V}_{h+1}(\cdot,\cdot) = \phi(\cdot,\cdot)^\top \widehat{w}_h$ is the solution of:

$$\widehat{w}_h = \arg\min_{w \in \mathbb{R}^d} \sum_{\tau \in \mathcal{D}} \left( r_h^\tau + \widehat{V}_{h+1}(x_{h+1}^\tau) - (\phi_h^\tau)^\top w \right)^2 + \lambda \|w\|_2^2 = \Lambda_h^{-1} \left( \sum_{\tau \in \mathcal{D}} \phi_h^\tau \big( r_h^\tau + \widehat{V}_{h+1}(x_{h+1}^\tau) \big) \right), \quad (2)$$

where $\Lambda_h = \sum_{\tau \in \mathcal{D}} \phi\left(x_h^\tau, a_h^\tau\right) \phi\left(x_h^\tau, a_h^\tau\right)^\top + \lambda I_d$. $\Gamma_h(x,a) = \beta \left\| \phi(x,a) \right\|_{\Lambda_h^{-1}}$ is a bonus function such that $|(\mathcal{T}_h \widehat{V}_{h+1} - \widehat{\mathcal{T}}_h \widehat{V}_{h+1})(x,a)| \leq \Gamma_h(x,a)$ for all $(x,a) \in \mathcal{S} \times \mathcal{A}$ with high probability. Intuitively, pessimism means that we use lower confidence bound by subtracting the bonus $\Gamma_h(\cdot,\cdot)$ to penalize the uncertainty so that $\widehat{Q}_h(x,a) \leq \mathcal{T}_h \widehat{V}_{h+1}(x,a)$. If pessimism is achieved for all steps $h \in [H]$, then we have the following key result (formally presented in Lemma 2):

$$V_1^*(x) - V_1^{\widehat{\pi}}(x) \leq 2 \sum_{h=1}^{H} \mathbb{E}_{\pi^*} \left[ \Gamma_h(x_h, a_h) | x_1 = x \right]. \quad (3)$$

Therefore, to establish a sharper suboptimality bound, it suffices to construct a smaller bonus function $\Gamma_h$ that can ensure pessimism. There remains, however, a gap of $\tilde{O}(\sqrt{d}H)$ between the suboptimality bound of PEVI and the minimax lower bound in Zanette et al. (2021).

**Self-normalized Process and Uniform Concentration**. Given a function $f_{h+1}$, to construct the bonus function, we can bound $|\mathcal{T}_h f_{h+1} - \widehat{\mathcal{T}}_h f_{h+1}|$ as follows (formally presented in Lemma 3):

$$\left| (\mathcal{T}_h f_{h+1})(x,a) - (\widehat{\mathcal{T}}_h f_{h+1})(x,a) \right| \lesssim \underbrace{\left\| \sum_{\tau \in \mathcal{D}} \phi\left(x_h^\tau, a_h^\tau\right) \cdot \xi_h^\tau(f_{h+1}) \right\|_{\Lambda_h^{-1}}}_{\text{(A)} \leq \beta} \left\| \phi(x,a) \right\|_{\Lambda_h^{-1}}, \quad (4)$$

where $\xi_h^\tau(f_{h+1}) := r_h^\tau + f_{h+1}(x_{h+1}^\tau) - (\mathcal{T}_h f_{h+1})(x_h^\tau, a_h^\tau)$. Bounding (A) is referred to as the concentration of a *self-normalized process* in the literature (Abbasi-Yadkori et al., 2011). For any *fixed* $\widehat{V}_{h+1}$, Lemma 9 ensures a high-probability upper bound $\tilde{O}(H\sqrt{d})$ for (A). However, in the backward iteration, $\widehat{V}_{h+1}$ is computed by ridge regression in later steps $[h+1, H]$ and thus inevitably depends on $\{x_{h+1}^\tau\}_{\tau \in \mathcal{D}}$, which is also used to estimate the Bellman equation at step $h$. Consequently, the concentration inequality cannot directly be applied since the martingale filtration in Lemma 9 is not well-defined. To resolve the above measurability issue, the standard approach is to establish a uniform concentration result over an $\epsilon$-covering of the following function class of $\widehat{V}_{h+1}$:

$$\mathcal{V}_{h+1} := \max_a \{ \phi(\cdot,a)^\top w + \beta \left\| \phi(\cdot,a) \right\|_{\Lambda_h^{-1}}, \|w\| \leq R, \beta \in [0,B], \Lambda \succeq \lambda \cdot I \}_{[0, H-h+1]}.$$

Then, the self-normalized process is bounded by the uniform bound of the $\epsilon$-covering, plus some approximation error. We can tune the parameter $\epsilon > 0$ and obtain that ((B.20) of Jin et al. (2021c)):

$$\left\| \sum_{\tau \in \mathcal{D}} \phi(x_h^\tau, a_h^\tau) \cdot \xi_h^\tau(\widehat{V}_{h+1}) \right\|_{\Lambda_h^{-1}}^2 \leq cH^2 \cdot \left( \log\left( H \mathcal{N}_{h+1}(\epsilon)/\delta \right) + d \log(1+K) + d^2 \right)$$

$$\leq cH^2 \cdot \big( \underbrace{d^2}_{\text{Uniform conver.}} + \underbrace{d}_{\text{Conver. + fixed } V} + \underbrace{d^2}_{\text{Approximation error}} \big),$$

where we use an upper bound of the covering number (Lemma 11), and omit all the logarithmic terms for a clear comparison. Therefore, we conclude that the uniform concentration leads to an extra $\sqrt{d}$ from the logarithmic covering number $\log \mathcal{N}_{h+1}(\epsilon)$. The prior work Yin et al. (2022) omits the dependency between $\widehat{V}_{h+1}$ and $\{x_h^\tau, a_h^\tau, x_{h+1}^\tau\}_{\tau \in \mathcal{D}}$, thus circumventing the uniform concentration. To fix this gap, one might make an additional assumption that the dataset is independent across different time steps $h$ as in Min et al. (2021), which is not realistic in practice because the behavior policy collects the trajectories by playing the episode starting from the initial state.

## 4 REFERENCE-ADVANTAGE DECOMPOSITION UNDER LINEAR FUNCTION APPROXIMATION

In this section, we introduce the reference-advantage decomposition under linear function approximation, which serves to avoid a $\sqrt{d}$ amplification of the error due to the uniform concentration.

We first define $\widehat{\mathbb{P}}_h g_{h+1}(\cdot, \cdot)$ to be an estimator of the conditional expectation, which is obtained by setting $r_h^\tau = 0$ in $\widehat{\mathcal{T}}_h g_{h+1}(\cdot, \cdot)$. We observe that the following affine properties hold:

$$
\left(\widehat{\mathcal{T}}_h(f_{h+1} + g_{h+1})\right)(\cdot, \cdot) = \phi(\cdot, \cdot)^\top \Lambda_h^{-1}\left(\sum_{\tau \in \mathcal{D}} \phi_h^\tau (r_h^\tau + f_{h+1}(x_{h+1}^\tau))\right) + \phi(\cdot, \cdot)^\top \Lambda_h^{-1}\left(\sum_{\tau \in \mathcal{D}} \phi_h^\tau g_{h+1}(x_{h+1}^\tau)\right)
$$

$$
= \widehat{\mathcal{T}}_h f_{h+1}(\cdot, \cdot) + \widehat{\mathbb{P}}_h g_{h+1}(\cdot, \cdot),
$$

$$
(\mathcal{T}_h (f_{h+1} + g_{h+1}))(\cdot, \cdot) = \langle \phi(x, a), \theta_h \rangle + \int_\mathcal{S} f_{h+1}(x') \langle \phi(x, a), d\mu_h(x') \rangle + \int_\mathcal{S} g_{h+1}(x') \langle \phi(x, a), d\mu_h(x') \rangle
$$

$$
= \mathcal{T}_h f_{h+1}(\cdot, \cdot) + \mathbb{P}_h g_{h+1}(\cdot, \cdot).
$$

Instead of directly bounding the uncertainty as in Eqn. (4), we now make the following decomposition:

$$
\widehat{\mathcal{T}}_h \widehat{V}_{h+1}(x, a) - \mathcal{T}_h \widehat{V}_{h+1}(x, a) = \left(\widehat{\mathcal{T}}_h(\widehat{V}_{h+1} + V_{h+1}^* - V_{h+1}^*)\right)(x, a) - \left(\mathcal{T}_h(\widehat{V}_{h+1} + V_{h+1}^* - V_{h+1}^*)\right)(x, a)
$$

$$
= \underbrace{\widehat{\mathcal{T}}_h V_{h+1}^*(x, a) - \mathcal{T}_h V_{h+1}^*(x, a)}_{\text{Reference uncertainty} \leq b_{0,h}(x, a)} + \underbrace{\widehat{\mathbb{P}}_h(\widehat{V}_{h+1} - V_{h+1}^*)(x, a) - \mathbb{P}_h(\widehat{V}_{h+1} - V_{h+1}^*)(x, a)}_{\text{Advantage uncertainty} \leq b_{1,h}(x, a)}.
$$

$$
(5)
$$

We have the following key observations about the reference part and the advantage part.

**Circumventing the Uniform Concentration for Reference Function.** As $V_{h+1}^*$ is deterministic, we can directly invoke standard concentration inequality (Lemma 3 and 9) to set $b_{0,h}(x, a) = \tilde{O}(\sqrt{d}H) \|\phi(x, a)\|_{\Lambda_h^{-1}}$, which avoids a $\sqrt{d}$ amplification due to uniform concentration.

**High-order Error from Correlated Advantage Function.** Although we still need the standard uniform concentration argument to analyze the advantage function and obtain a sub-optimal $d$-dependency, under Assumption 1, by a carefully-crafted induction procedure, we can show that $\|\widehat{V}_{h+1} - V_{h+1}^*\|_\infty = \tilde{O}(\frac{\sqrt{d}H^2}{\sqrt{K\kappa}})$. The much smaller range implies that we can invoke Lemma 9 to set $b_{1,h}(x, a) = \tilde{O}(\frac{d^{3/2}H^2}{\sqrt{K\kappa}}) \|\phi(x, a)\|_{\Lambda_h^{-1}}$, which is non-dominating when $K \geq \tilde{\Omega}\left(d^2 H^2/\kappa\right)$. The detailed proof can be found in Appendix D.

Based on the above reasoning, we conclude that we can set $\Gamma_h(\cdot, \cdot) = \tilde{O}\left(\sqrt{d}H\right) \|\phi(x, a)\|_{\Lambda_h^{-1}}$ in the original PEVI algorithm (Jin et al., 2021c) and obtain a new algorithm referred to as LinPEVI-ADV. Together with a refined analysis, we have the following theoretical guarantee.

**Theorem 1** (LinPEVI-ADV). *Under the Assumptions 1, and $K > \tilde{\Omega}(d^2 H^2/\kappa)$, if we set $\lambda = 1$ and $\beta_1 = \tilde{O}(\sqrt{d}H)$ in Algorithm 2, then, with probability at least $1 - \delta$, for any $x \in \mathcal{S}$, we have*

$$
V_1^*(x) - V_1^{\widehat{\pi}}(x) \leq \tilde{O}\left(\sqrt{d}H\right) \sum_{h=1}^H \mathbb{E}_{\pi^*}\left[\|\phi(x, a)\|_{\Lambda_h^{-1}} \mid x_1 = x\right],
$$

*where $\Lambda_h = \sum_{\tau \in \mathcal{D}} \phi(x_h^\tau, a_h^\tau)\phi(x_h^\tau, a_h^\tau)^\top + \lambda I_d$.*

We remark that LinPEVI-ADV shares the same pseudo code with PEVI, except for a $\sqrt{d}$-improvement in the choice of $\beta$. For completeness, we present the code in Algorithm 2.

**Compared to Xie et al. (2021b).** To the best of our knowledge, the reference-advantage decomposition is new in the literature of offline linear MDP, but it is relatively well-studied in tabular MDPs (Azar et al., 2017; Zanette and Brunskill, 2019; Zhang et al., 2020; Xie et al., 2021b). Among them, Xie et al. (2021b) studies the offline tabular MDP and is most related to ours. While we share similar algorithmic ideas in terms of uncertainty decomposition, we comment on our differences as follows. First, in terms of algorithmic design, Xie et al. (2021b) uses an independent dataset to explicitly construct a reference $\widehat{V}_{h+1}^{\text{ref}}$ but we only use $V_{h+1}^*$ in the theoretical analysis. Second, in terms of the

---

**Algorithm 1** LinPEVI-ADV+

---

1: **Initialize:** Input datasets $\mathcal{D}$, $\mathcal{D}'$, and $\beta_2$; $\widehat{V}_{H+1}(\cdot) = 0$.
2: Construct variance estimator $\widehat{\sigma}_h^2(x_h^\tau, a_h^\tau)$ as in Section 5 with $\mathcal{D}'$.
3: **for** $h = H, \ldots, 1$ **do**
4:     $\Sigma_h = \sum_{\tau \in \mathcal{D}} \phi(x_h^\tau, a_h^\tau) \phi(x_h^\tau, a_h^\tau)^\top / \widehat{\sigma}_h^2(x_h^\tau, a_h^\tau) + \lambda I_d$;
5:     $\widehat{w}_h = \Sigma_h^{-1} \left( \sum_{\tau \in \mathcal{D}} \phi(x_h^\tau, a_h^\tau) \frac{r_h^\tau + \widehat{V}_{h+1}(x_{h+1}^\tau)}{\widehat{\sigma}_h^2(x_h^\tau, a_h^\tau)} \right)$;
6:     $\Gamma_h(\cdot, \cdot) \leftarrow \beta_2 \|\phi(\cdot, \cdot)\|_{\Sigma_h^{-1}}$;
7:     $\widehat{Q}_h(\cdot, \cdot) \leftarrow \{\phi(\cdot, \cdot)^\top \widehat{w}_h - \Gamma_h(\cdot, \cdot)\}_{[0, H-h+1]}$;
8:     $\widehat{\pi}_h(\cdot \mid \cdot) \leftarrow \arg\max_{\pi_h} \langle \widehat{Q}_h(\cdot, \cdot), \pi_h(\cdot \mid \cdot) \rangle_{\mathcal{A}}, \widehat{V}_h(\cdot) \leftarrow \langle \widehat{Q}_h(\cdot, \cdot), \widehat{\pi}_h(\cdot \mid \cdot) \rangle_{\mathcal{A}}$.
9: **end for**
10: **Output:** $\widehat{\pi} = \{\widehat{\pi}_h\}_{h=1}^H$.

---

uncertainty estimation, Xie et al. (2021b) estimates the uncertainty separately of each state-action pair by counting its frequency. On the contrary, in the linear setting, we deal with the regression and the state-action pairs are coupled with each other because the analysis of the self-normalized process involves the estimated covariance matrix thus all the samples at step $h$ (4). This brings distinct challenges to the linear setting, both in terms of the analysis and the coverage condition for achieving a sharp bound. Finally, the theoretical analyses are distinctly different due to the previous two points. For instance, we adopt a carefully-crafted induction procedure to control the advantage function, while Xie et al. (2021b) further introduces some dataset splitting techniques to handle the temporal dependency in the advantage function. Moreover, for linear MDP, the way in which the variance is introduced is also different (see Section 5).

## 5 LEVERAGE THE VARIANCE INFORMATION

The variance-weighted ridge regression technique was introduced by Zhou et al. (2021) in the literature of RL and was later firstly adapted by Min et al. (2021); Yin et al. (2022) to offline RL under linear function approximation. Specifically, given a *fixed* function $f_{h+1} : \mathcal{S} \to [0, H-1]$ as the target, we perform the following *variance-weighted* ridge regression to estimate $\widehat{w}_h$ as:

$$\operatorname*{argmin}_{w \in \mathbb{R}^d} \sum_{\tau \in \mathcal{D}} \frac{\left[(\phi_h^\tau)^\top w - r_h^\tau - f_{h+1}(x_{h+1}^\tau)\right]^2}{\widehat{\sigma}_h^2(x_h^\tau, a_h^\tau)} + \lambda \|w\|_2^2 = \Sigma_h^{-1} \left( \sum_{\tau \in \mathcal{D}} \frac{\phi(x_h^\tau, a_h^\tau) \cdot (r_h^\tau + f_{h+1}(x_{h+1}^\tau))}{\widehat{\sigma}_h^2(x_h^\tau, a_h^\tau)} \right),$$
(6)

where $\Sigma_h = \sum_{\tau \in \mathcal{D}} \frac{\phi(x_h^\tau, a_h^\tau) \phi(x_h^\tau, a_h^\tau)^\top}{\widehat{\sigma}_h^2(x_h^\tau, a_h^\tau)} + \lambda I_d$ and $\widehat{\sigma}_h^2(\cdot, \cdot) \in [1, H^2]$ is an *independent* variance estimator. In this case, we can bound the uncertainty by Lemma 3 as follows:

$$|\mathcal{T}_h f_{h+1} - \widehat{\mathcal{T}}_h f_{h+1}|(x, a) \lesssim \| \sum_{\tau \in \mathcal{D}} \phi(x_h^\tau, a_h^\tau) \cdot \xi_h^\tau(f_{h+1}) \|_{\Sigma_h^{-1}} \cdot \|\phi(x, a)\|_{\Sigma_h^{-1}},$$

with $\xi_h^\tau(f_{h+1}) = \frac{r_h^\tau + f_{h+1}(x_{h+1}^\tau) - (\mathcal{T}_h f_{h+1})(x_h^\tau, a_h^\tau)}{\widehat{\sigma}_h(x_h^\tau, a_h^\tau)}$. The high-level idea is that the normalized $\xi_h^\tau(f_{h+1})$ is of a conditional variance $O(1)$ and the Bernstein-type inequality (Lemma 10) gives a bound of $\beta_2 := \tilde{O}(\sqrt{d})$ for the self-normalized process instead of $\beta_1 = \tilde{O}(H\sqrt{d})$ from the Hoeffding-type one. Therefore, instead of depending on the planning horizon $H$ explicitly, for $\beta_2 \|\phi(x, a)\|_{\Sigma_h^{-1}}$, the $H$ factor is "hidden" in the covariance matrix $\Sigma_h$. Furthermore, as $\Sigma_h^{-1} \preccurlyeq H^2 \Lambda_h^{-1}$, we know that the new bonus function is never worse because $\beta_2 \|\phi(x, a)\|_{\Sigma_h^{-1}} \lesssim \beta_1 \|\phi(x, a)\|_{\Lambda_h^{-1}}$. Combining this observation with (3), we conclude that the PEVI with variance-weighted regression is superior as long as we can construct an *independent* and approximately accurate variance estimator $\widehat{\sigma}_h^2(\cdot, \cdot)$. To this end, we also carefully handle the temporal dependency due to the following two reasons: (i) we need to avoid the measurability issue as described in Section 4; and (ii) the estimation of the conditional variance of $\xi_h^\tau(f_{h+1})$ will be hard otherwise. To further illustrate the idea, we now discuss the limitations of existing approaches when we do not omit the temporal dependency, thus motivating our corresponding modifications.

**Limitation of the Existing Approach**. Yin et al. (2022) constructs the variance estimator $\widehat{\sigma}_h^2(\cdot, \cdot)$ and estimate the Bellman equation both with the target function $\widehat{V}_{h+1}$. As discussed in Section 3, $\widehat{\sigma}_h^2(\cdot, \cdot)$ is statistically dependent with the dataset $\mathcal{D}_h := \{(x_h^\tau, a_h^\tau)\}_{\tau=1}^K$ due to temporal dependency. Therefore, the random variables $\{\frac{\phi(x_h^\tau, a_h^\tau)}{\widehat{\sigma}_h(x_h^\tau, a_h^\tau)}\}_{\tau=1}^K$ are dependent so the concentration of the covariance matrix with Lemma 12 (Lemma C.3 of Yin et al. (2022)) does not apply. Moreover, a similar measurability issue arises from the statistical dependency between the $\widehat{\sigma}_h(\cdot, \cdot)$ and $\mathcal{D}_h$ so the concentration of the self-normalized process also fails. Finally, the conditional variance of $\widehat{V}_{h+1}(x_{h+1}^\tau)/\widehat{\sigma}_h(x_h^\tau, a_h^\tau)$ is hard to control because the numerator and denominator are tightly coupled with each other.

**Variance Estimator**. Equipped with the reference-advantage decomposition, it suffices to focus on the reference function $V_{h+1}^*$. If we can construct a $\widehat{\sigma}_h^2(\cdot, \cdot)$ that is independent of $\mathcal{D}$, then $\xi_h^\tau(V_{h+1}^*) = \frac{r_h^\tau + V_{h+1}^*(x_{h+1}^\tau) - (\mathcal{T}_h V_{h+1}^*)(x_h^\tau, a_h^\tau)}{\widehat{\sigma}_h(x_h^\tau, a_h^\tau)}$ will be independent with each other and easy to deal with. To this end, we first use an independent dataset $\mathcal{D}'$ to run Algorithm 2 and construct $\{\widehat{V}_h'\}_{h=1}^H$ (this only incurs a factor of 2 in the final sample complexity). Then, by Lemma 1, we know that there exist $\beta_{h,1}, \beta_{h,2} \in \mathbb{R}^d$ such that $[\mathbb{P}_h(\widehat{V}_{h+1}')^2](x, a) = \langle \phi(x, a), \beta_{h,2} \rangle$ and $([\mathbb{P}_h \widehat{V}_{h+1}'(x, a)])^2 = [\langle \phi(x, a), \beta_{h,1} \rangle]^2$. Similarly, we can approximate $\beta_{1,h}$ and $\beta_{2,h}$ via ridge regression:

$$
\begin{aligned}
\widetilde{\beta}_{h,2} &= \operatorname*{argmin}_{\beta \in \mathbb{R}^d} \sum_{\tau \in \mathcal{D}'} \left[ \langle \phi(x_h^\tau, a_h^\tau), \beta \rangle - (\widehat{V}_{h+1}')^2 (x_{h+1}^\tau) \right]^2 + \lambda \|\beta\|_2^2, \\
\widetilde{\beta}_{h,1} &= \operatorname*{argmin}_{\beta \in \mathbb{R}^d} \sum_{\tau \in \mathcal{D}'} \left[ \langle \phi(x_h^\tau, a_h^\tau), \beta \rangle - \widehat{V}_{h+1}'(x_{h+1}^\tau) \right]^2 + \lambda \|\beta\|_2^2.
\end{aligned}
\tag{7}
$$

We then employ the following variance estimator:

$$
\widehat{\sigma}_h^2(x, a) := \max \left\{ 1, \left[ \phi(x, a)^\top \widetilde{\beta}_{h,2} \right]_{[0, H^2]} - \left[ \phi(x, a)^\top, \widetilde{\beta}_{h,1} \right]_{[0, H]}^2 - \tilde{O}\left( \frac{dH^3}{\sqrt{K\kappa}} \right) \right\}.
\tag{8}
$$

With proof essentially similar to that of PEVI, we can show that with high probability (cf. Lemma 5),

$$
[\mathbb{V}_h V_{h+1}^*](x, a) - \tilde{O}\left( \frac{dH^3}{\sqrt{K\kappa}} \right) \le \widehat{\sigma}_h^2(x, a) \le [\mathbb{V}_h V_{h+1}^*](x, a),
$$

where $[\mathbb{V}_h V_{h+1}^*](x, a) = \max\{1, [\operatorname{Var}_h V_{h+1}^*](x, a)\}$ is the truncated variance of $V_{h+1}^*(\cdot)$. Therefore, $\widehat{\sigma}_h^2(\cdot, \cdot)$ defined in (8) approximates the conditional variance of $\xi_h^\tau(V_{h+1}^*)$ well when $K$ exceeds a threshold. Moreover, since the target function $V_{h+1}^*$ is deterministic thus measurable, we know that

$$
\begin{aligned}
\operatorname{Var}_{x_{h+1}^\tau | x_h^{1:\tau}, a_h^{1:\tau}, x_{h+1}^{1:\tau-1}} \left[ \frac{V_{h+1}^*(x_{h+1}^\tau)}{\widehat{\sigma}_h(x_h^\tau, a_h^\tau)} \right] &= \operatorname{Var}_{x_{h+1}^\tau | x_h^\tau, a_h^\tau} \left[ \frac{V_{h+1}^*(x_{h+1}^\tau)}{\widehat{\sigma}_h(x_h^\tau, a_h^\tau)} \right] \\
&= \frac{\operatorname{Var}_{x_{h+1}^\tau | x_h^\tau, a_h^\tau}[V_{h+1}^*(x_{h+1}^\tau)]}{\widehat{\sigma}_h(x_h^\tau, a_h^\tau)} \approx O(1),
\end{aligned}
$$

where $x_h^{1:\tau}, a_h^{1:\tau}, x_{h+1}^{1:\tau-1}$ is short for $\{(x_h^i, a_h^i) : 1 \le i \le \tau\} \cup \{x_{h+1}^i : 1 \le i \le \tau - 1\}$. Therefore, the high-level idea stated at the beginning of this section is realized by the variance estimator defined by (8). Combining the variance-weighted regression with the reference-advantage decomposition, we propose LinPEVI-ADV+ (Algorithm 1), which enjoys the following theoretical guarantee.

**Theorem 2** (LinPEVI-ADV+). *Under Assumption 1, for $K \ge \tilde{\Omega}(d^2 H^6/\kappa)$, if we set $\lambda = 1/H^2$ and $\beta_2 = \tilde{O}(\sqrt{d})$ in Algorithm 1, then, we have with probability at least $1 - \delta$, for any $x \in \mathcal{S}$, we have*

$$
V_1^*(x) - V_1^{\widehat{\pi}}(x) \le \tilde{O}(\sqrt{d}) \cdot \sum_{h=1}^H \mathbb{E}_{\pi^*} \left[ \|\phi(x_h, a_h)\|_{\Sigma_h^{*-1}} \mid x_1 = x \right],
$$

*where $\Sigma_h^* = \sum_{\tau \in \mathcal{D}} \phi(x_h^\tau, a_h^\tau)\phi(x_h^\tau, a_h^\tau)^\top / [\mathbb{V}_h V_{h+1}^*](x_h^\tau, a_h^\tau) + \lambda I_d$.*

**Interpretation of the result.** Since $[\mathbb{V}_h V_{h+1}^*](\cdot, \cdot) \in [1, H^2]$, we know that $\Sigma_h^{*-1} \preccurlyeq H^2 \Lambda_h^{-1}$. This implies that LinPEVI-ADV+ is never worse than LinPEVI-ADV thus also superior to the PEVI. Such an improvement is indeed strict when it reduces to the tabular setting. Specifically, let $d_h^*(x, a)$ denote the distribution of visitation of $(x, a)$ at step $h$ under the optimal policy $\pi^*$. LinPEVI-ADV gives a

| | Independence Assumption | Suboptimality Bound |
|---|---|---|
| Jin et al. (2021c) | ✗ | $\tilde{O}(dH) \cdot \sum_{h=1}^{H} \mathbb{E}_{\pi^*}[\|\phi(x,a)\|_{\Lambda_h^{-1}} \mid x_1 = x]$ |
| Yin et al. (2022) | ✓ | $\tilde{O}(\sqrt{d}) \cdot \sum_{h=1}^{H} \mathbb{E}_{\pi^*}[\|\phi(x_h,a_h)\|_{\Sigma_h^{*-1}} \mid x_1 = x]$ |
| Theorem 1 | ✗ | $\tilde{O}(\sqrt{d}H) \cdot \sum_{h=1}^{H} \mathbb{E}_{\pi^*}[\|\phi(x,a)\|_{\Lambda_h^{-1}} \mid x_1 = x]$ |
| Theorem 2 | ✗ | $\tilde{O}(\sqrt{d}) \cdot \sum_{h=1}^{H} \mathbb{E}_{\pi^*}[\|\phi(x_h,a_h)\|_{\Sigma_h^{*-1}} \mid x_1 = x]$ |

Table 1: A comparison with existing results. Here $\Lambda_h = \sum_{\tau \in \mathcal{D}} \phi(x_h^\tau, a_h^\tau)\phi(x_h^\tau, a_h^\tau)^\top + \lambda I_d$ and $\Sigma_h^* = \sum_{\tau \in \mathcal{D}} \phi(x_h^\tau, a_h^\tau)\phi(x_h^\tau, a_h^\tau)^\top / [\mathbb{V}_h V_{h+1}^*](x_h^\tau, a_h^\tau) + \lambda I_d$. The independence assumption represents the assumption that the data samples are independent across different time steps $h$. We also remark that, in the new version of Jin et al. (2021c), they adopt a dataset splitting technique to split the dataset into $H$ independent subsets. This technique essentially shares the same spirit with the assumption and is at the expense of an additional $H$ factor in the final sample complexity.

bound $\tilde{O}(\sqrt{d}H \sum_{h,x,a} d_h^*(x,a)\sqrt{\frac{1}{Kd_h^b(x,a)}})$ which has a horizon dependence of $H^2$. On the other hand, LinPEVI-ADV+ gives a bound of the form $\tilde{O}(\sqrt{d} \sum_{h,x,a} d_h^*(x,a)\left[\frac{[\mathbb{V}_h V_{h+1}^*](x,a)}{Kd_h^b(x,a)}\right]^{1/2})$, which has a horizon dependence of $H^{3/2}$ by law of total variance. Meanwhile, LinPEVI-ADV+ enjoys the same rate as the VAPVI (Yin et al., 2022) without an additional independence assumption on the offline dataset, as summarized in the Table 1.

**Compared to other methods.** Zanette et al. (2020) and Xie et al. (2021a) also achieve a $\sqrt{d}$ improvement but the algorithmic ideas are fundamentally different from ours. In specific, the actor-critic-based Zanette et al. (2020) establishes pessimism via direct perturbations of the parameter vectors[2]. Xie et al. (2021a) establishes pessimism only at the initial value and is only information-theoretic. We develop techniques to resolve the issue in the LSVI framework because it possesses an appealing feature that we can assign sample-dependent weights in the regression thus obtaining a sharp $H$-dependence. To the best of our knowledge, no similar result is available in the other two frameworks. When rescaling the range of V-value to $[0, H]$, their bounds will be sub-optimal. Moreover, their bounds depend on the cardinality of the action space, while the LSVI-based one can deal with an infinite action space.

To further interpret our result, we state the following lower bound for offline linear MDP.

**Theorem 3** (Lower Bound for MDP). *For fixed episode length $H$, dimension $d$, probability $\delta$, and sample size $K \geq \tilde{\Omega}(d^4)$. There exists a class $\mathcal{M}$ of linear MDPs and an offline dataset $\mathcal{D}$ with $|\mathcal{D}| = K$, such that for any policy $\hat{\pi}$, it holds with probability at least $1 - \delta$ that for some universal constant $c > 0$, we have*

$$\sup_{M \in \mathcal{M}} \mathbb{E}_M[V_1^*(x_1) - V_1^{\hat{\pi}}(x_1)] \geq c\sqrt{d} \cdot \sum_{h=1}^{H} \mathbb{E}_{\pi^*}\|\phi(x_h, a_h)\|_{\Sigma_h^{*-1}}.$$

The lower bound matches Theorem 2, thus establishing the optimality of LinPEVI-ADV+ for sufficiently large $K$. We also remark that Yin et al. (2022) establishes the lower bound $c\sqrt{d} \cdot \sum_{h=1}^{H} \|\mathbb{E}_{\pi^*}\phi(x_h, a_h)\|_{\Sigma_h^{*-1}}$, which is smaller than our lower bound due to Jensen's inequality.

## 6 EXTENSIONS

### 6.1 LINEAR MDP WITH FINITE FEATURE SET

As shown in the seminal work of linear contextual bandit Chu et al. (2011), one can further improve the suboptimality bound by a factor of $\sqrt{d}$, when the action set is finite. Equipped with the technique developed above, we can obtain a similar improvement in the case of finite features.

**Assumption 2** (Finite Feature Set). *We assume that $|\{\phi(x,a) \in \mathbb{R}^d : x \in \mathcal{S}, a \in \mathcal{A}\}| = M < \infty$.*

---

[2]See page 17 of Zanette et al. (2020) for a discussion about the error amplification issue.

We start with bounding $|\mathcal{T}_h V_{h+1}^* - \widehat{\mathcal{T}}_h V_{h+1}^*|$, which is the key to establishing the bonus function:

$$\left(\widehat{\mathcal{T}}_h V_{h+1}^* - \mathcal{T}_h V_{h+1}^*\right)(x,a) \overset{(a)}{\lesssim} \sum_{\tau \in \mathcal{D}} \underbrace{\left\langle \phi(x,a), \Lambda_h^{-1} \phi_h^\tau \right\rangle}_{\text{constant}} \xi_h^\tau(V_{h+1}^*) \overset{(b)}{\leq} \|\phi(x,a)\|_{\Lambda_h^{-1}} \left\| \sum_{\tau \in \mathcal{D}} \xi_h^\tau(V_{h+1}^*) \phi_h^\tau \right\|_{\Lambda_h^{-1}},$$

where we prove (a) in Appendix I and (b) follows from Cauchy-Schwarz inequality. The reasoning proceeds as follows. The key observation is that since $V_{h+1}^*$ is deterministic, $\{\xi_h^\tau(V_{h+1}^*)\}_{\tau \in \mathcal{D}}$ are independent conditioned on $\mathcal{D}_h = \{(x_h^\tau, a_h^\tau)\}_{\tau \in \mathcal{D}}$ thus the classic Hoeffding's inequality applies. To get a high-probability bound for all features, Assumption 2 allows us to bound the middle term directly by paying for a $\log(M)$ from a union bound argument, instead of a $\sqrt{d}$ from Lemma 9. We remark that the decomposition trick is necessary for the above reasoning. One cannot take condition on $\mathcal{D}_h$ otherwise because this will influence the distribution of $\{\xi_h^\tau(\widehat{V}_{h+1})\}_{\tau \in \mathcal{D}}$. Combining this with (3), we have the following result.

**Theorem 4** (LinPEVI-ADV with Finite Feature Set). *Suppose Assumptions 1 and 2 hold. If $K \geq \tilde{\Omega}(d^2 H^2/\kappa)$ and we set $\beta_1 = O\left(\sqrt{\log(2H^2M/\delta)}\right)$ and $\lambda = 1/d$ in Algorithm 2, then, with probability at least $1 - \delta$, for any $x \in \mathcal{S}$, with $\Lambda_h = \sum_{\tau \in \mathcal{D}} \phi(x_h^\tau, a_h^\tau)\phi(x_h^\tau, a_h^\tau)^\top + \lambda I_d$, we have*

$$V_1^*(x) - V_1^{\widehat{\pi}}(x) \leq O\left(H\left[\log\left(2H^2M/\delta\right)\right]^{1/2}\right) \sum_{h=1}^H \mathbb{E}_{\pi^*}\left[\|\phi(x_h, a_h)\|_{\Lambda_h^{-1}} \mid x_1 = x\right].$$

## 6.2 LINEAR TWO-PLAYER ZERO-SUM MARKOV GAME

In the two-player zero-sum MGs (Xie et al., 2020) where at each step, there is another player taking action simultaneously from another action space $\mathcal{B}$ and the reward function and transition kernel are linear in a feature map $\phi(x,a,b) : \mathcal{S} \times \mathcal{A} \times \mathcal{B} \to \mathbb{R}^d$. The learning objective is to approximate the Nash equilibrium (NE): $(\pi^*, \nu^*)$ such that $V_h^{\pi^*, \nu^*}(x) = \max_\pi \min_\nu V_h^{\pi, \nu}(x)$, where the V-value function is now defined as $V_h^{\pi, \nu}(x) = \mathbb{E}_{\pi, \nu}[\sum_{h'=h}^H r_{h'} | x_h = x]$. With slight abuse of notation, we can define the Bellman operator for the two-player zero-sum MG as follows:

$$(\mathcal{T}_h V)(x,a,b) := r_h(x,a,b) + \sum_{x' \in \mathcal{S}} \mathbb{P}_h(x'|x,a,b)V(x') = \phi(x,a,b)^\top w_h, \tag{9}$$

where the linear structure of the Bellman equation (i.e. the existence of $w_h \in \mathbb{R}^d$) follows from the linearity of reward and transition. The Pessimistic Minimax Value Iteration (PMVI) proposed in Zhong et al. (2022) also establishes pessimism at every step and we have

$$\underbrace{V_1^*(x) - \min_\nu V_1^{\widehat{\pi}, \nu}(x)}_{\text{Gap to the NE Value}} \leq 2 \sup_\pi \sum_{h=1}^H \mathbb{E}_{\pi, \nu^*}[\underline{\Gamma}_h(x,a,b) \mid x_1 = x], \tag{10}$$

where $\underline{\Gamma}_h(x,a,b)$ is a bonus function such that $|\mathcal{T}_h \underline{V}_{h+1}(x,a,b) - \widehat{\mathcal{T}}_h \underline{V}_{h+1}(x,a,b)| \leq \underline{\Gamma}_h(x,a,b)$ with high probability and $\underline{V}_{h+1}$ is the estimated NE value of the first agent at step $h + 1$. Although the learning objective is different, the suboptimality bound essentially also reduces to the uncertainty estimation at each step, and Zhong et al. (2022) suffers from exactly the same challenge from the statistical dependency between $\underline{V}_{h+1}$ and the data samples used to construct $(\widehat{\mathcal{T}}_h \underline{V}_{h+1})$. Therefore, our techniques can be readily extended to this the MG setting and improve the result in Zhong et al. (2022). We defer the details to Appendix B.

## 7 CONCLUSION

In this paper, we study the linear MDPs in the offline setting. We identify the complicated statistical dependency between different time steps as the bottleneck of the algorithmic design and theoretical analysis. To address this issue, we develop a new reference-advantage decomposition technique under the linear function approximation, which serves to avoid a $\sqrt{d}$-amplification of the value function error due to temporal dependency and is also critical for leveraging the variance information to achieve a sharp dependence on the planning horizon $H$. We further generalize the developed techniques to the linear MDP with finite features and also the two-player zero-sum MGs, which demonstrate the broad adaptability of our methods.

ACKNOWLEDGEMENTS

Wei Xiong and Tong Zhang acknowledge the funding supported by the GRF 16310222 GRF 16201320. Chengshuai Shi and Cong Shen acknowledge the funding support by the US National Science Foundation under Grant ECCS-2029978, ECCS-2033671, ECCS-2143559, CNS-2002902, and the Bloomberg Data Science Ph.D. Fellowship. Liwei Wang is supported by National Key R&D Program of China (2022ZD0114900) and National Science Foundation of China (NSFC62276005).

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

# A    NOTATION TABLE AND COMPARISONS

## A.1    NOTATIONS TABLE

We summarize the notations used in this paper in the Table 2.

| Notation | Explanation |
|---|---|
| $\kappa = \min_{h\in[H]} \lambda_{\min}(\mathbb{E}_{d_h^b}[\phi(x,a)\phi(x,a)^\top]) > 0$ | Assumption 1 (MDP) and 3 (MG) |
| $(\mathbb{P}_h V)(x,a) = \sum_{x'\in\mathcal{S}} \mathbb{P}_h(x'|x,a)V(x')$ | conditional expectation |
| $[\mathrm{Var}_h V](x,a) = [\mathbb{P}_h V^2](x,a) - ([\mathbb{P}_h V](x,a))^2$ | conditional variance |
| $(\mathcal{T}_h V)(x,a) = r_h(x,a) + (\mathbb{P}_h V)(x,a)$ | Bellman equation and Bellman operator |
| $\widehat{\sigma}_h^2(\cdot,\cdot) \in [1, H^2]$ | empirical variance estimator |
| $[\mathbb{V}_h V_{h+1}^*](x,a) = \max\{1, [\mathrm{Var}_h V_{h+1}^*](x,a)\}$ | clipped conditional variance of $V_{h+1}^*$ |
| $\Lambda_h = \sum_{\tau\in\mathcal{D}} \phi(x_h^\tau, a_h^\tau)\phi(x_h^\tau, a_h^\tau)^\top + \lambda I_d$ | regular covariance estimator |
| $\Sigma_h = \sum_{\tau\in\mathcal{D}} \frac{\phi(x_h^\tau,a_h^\tau)\phi(x_h^\tau,a_h^\tau)^\top}{\widehat{\sigma}_h^2(x_h^\tau,a_h^\tau)} + \lambda I_d$ | variance-weighted covariance estimator |
| $\Sigma_h^* = \sum_{\tau\in\mathcal{D}} \frac{\phi(x_h^\tau,a_h^\tau)\phi(x_h^\tau,a_h^\tau)^\top}{[\mathbb{V}_h V_{h+1}^*](x_h^\tau,a_h^\tau)} + \lambda I_d$ | variance-weighted covariance matrix |
| $\xi_h^\tau(f_{h+1}) = \frac{r_h^\tau + f_{h+1}(x_{h+1}^\tau) - (\mathcal{T}_h f_{h+1})(x_h^\tau,a_h^\tau)}{\widehat{\sigma}_h(x_h^\tau,a_h^\tau)}$ | noise in the self-normalized process |

Table 2: A summary of notations used in this paper. With slight abuse of notations, the notations for MGs are defined similarly but we replace $(x,a)$ with $(x,a,b)$ and replace $(x_h^\tau, a_h^\tau)$ with $(x_h^\tau, a_h^\tau, b_h^\tau)$, accordingly.

## A.2    ADDITIONAL RELATED WORK

We review existing works that are closely related to our paper in this section.

**Offline RL.** The principle of pessimism is first used by Jin et al. (2021c) to empower efficient offline learning under only partial coverage. It shows that we can design an efficient offline RL algorithm with only sufficient coverage over the optimal policy, instead of the previous uniform one required by Precup (2000); Antos et al. (2008); Levine et al. (2020). After that, a line of work (Rashidinejad et al., 2021; Yin and Wang, 2021; Uehara et al., 2021; Zanette et al., 2021; Xie et al., 2021a; Uehara and Sun, 2021; Shi et al., 2022; Li et al., 2022) leverages the principle of pessimism, either in the tabular case or in the case with function approximation, and we elaborate them separately.

**Offline tabular RL.** For tabular MDP, a line of works has incorporated the principle of pessimism to design efficient offline RL algorithms (Rashidinejad et al., 2021; Yin and Wang, 2021; Xie et al., 2021b; Shi et al., 2022; Li et al., 2022). In particular, Xie et al. (2021b) proposes a variance-reduction offline RL algorithm for tabular MDP which is nearly optimal after the total sample size exceeds a certain threshold. After that, Li et al. (2022) proposed an algorithm that is nearly optimal by introducing a novel subsampling trick to cancel the temporal dependency among time steps. Shi et al. (2022) proposed the first nearly optimal model-free offline RL algorithm.

**Offline RL with function approximation.** For RL problems with linear function approximation, Jin et al. (2021c) designs the first pessimism-based efficient offline algorithm for linear MDP. After that, Min et al. (2021) considers offline policy evaluation problem under linear MDP and designs a novel offline algorithm which incorporates the variance information of the value function to improve the sample efficiency. This technique is later adopted by Yin et al. (2022). However, Min et al. (2021); Yin et al. (2022) depend (explicitly or implicitly) on an assumption that the data samples are independent across different time steps $h$ so they do not need to handle the temporal dependency, which considerably complicates the analysis. Moreover, such an assumption is not very realistic when the dataset is collected by some behavior policy by interacting with the underlying MDP. Therefore, it remains open whether we can design computationally efficient algorithms that achieve minimax optimal sample efficiency for offline learning with linear MDP. Beyond the linear function approximation, Xie et al. (2021a); Uehara and Sun (2021) propose pessimistic offline RL algorithms with general function approximation. However, their works are only information-theoretic as they require an optimization subroutine over the general function class which is computationally intractable in general.

---

**Algorithm 2** PEVI (LinPEVI-ADV)

---

1: **Initialize:** Input dataset $\mathcal{D}$, $\beta_1$; $\widehat{V}_{H+1}(\cdot) = 0$.
2: **for** $h = H, \ldots, 1$ **do**
3: $\quad \Lambda_h \leftarrow \sum_{\tau \in \mathcal{D}} \phi(x_h^\tau, a_h^\tau) \phi(x_h^\tau, a_h^\tau)^\top + \lambda I_d$;
4: $\quad \widehat{w}_h \leftarrow \Lambda_h^{-1} (\sum_{\tau \in \mathcal{D}} \phi(x_h^\tau, a_h^\tau)(r_h^\tau + \widehat{V}_{h+1}(x_{h+1}^\tau)))$;
5: $\quad \Gamma_h(\cdot, \cdot) \leftarrow \beta_1 \|\phi(\cdot, \cdot)\|_{\Lambda_h^{-1}}$;
6: $\quad \widehat{Q}_h(\cdot, \cdot) \leftarrow \{\phi(\cdot, \cdot)^\top \widehat{w}_h - \Gamma_h(\cdot, \cdot)\}_{[0, H-h+1]}$;
7: $\quad \widehat{\pi}_h(\cdot \mid \cdot) \leftarrow \arg\max_{\pi_h} \langle \widehat{Q}_h(\cdot, \cdot), \pi_h(\cdot \mid \cdot) \rangle_{\mathcal{A}}, \widehat{V}_h(\cdot) \leftarrow \langle \widehat{Q}_h(\cdot, \cdot), \widehat{\pi}_h(\cdot \mid \cdot) \rangle_{\mathcal{A}}$.
8: **end for**
9: **Output:** $\widehat{\pi} = \{\widehat{\pi}_h\}_{h=1}^H$ and $\widehat{V} = \{\widehat{V}_h\}_{h=1}^H$

---

**Offline MGs.** The existing works studying sample-efficient equilibrium finding in offline MARL include Zhong et al. (2021); Chen et al. (2021); Cui and Du (2022); Zhong et al. (2022). Among these works, Cui and Du (2022) and Zhong et al. (2022) are most closely related to our algorithm where they study the offline two-player zero-sum MGs in the tabular and linear case, respectively. In terms of the minimal dataset coverage assumption, both of them identify that the unilateral concentration, i.e., a good coverage on the $\{(\pi^*, \nu), (\pi, \nu^*) : (\pi^*, \nu^*) \text{ is an NE and } (\pi, \nu) \text{ is arbitrary.}\}$, is necessary and sufficient for efficient offline learning. In terms of sample complexity, similarly, while for the tabular game, Cui and Du (2022) is nearly optimal in terms of the dependency on the number of states, there is a $\tilde{O}(\sqrt{d}H)$ gap between the upper and lower bounds for linear MG (Zhong et al., 2022).

**Online RL with function approximation.** Jin et al. (2020) and Xie et al. (2020) propose the first provably efficient algorithm for online linear MDP and linear MG, respectively. However, there is a gap between their regret bounds and existing lower bounds where we remark that similar issues of temporal dependency also exist in the analysis of online algorithms for linear MDP and linear MG. For instance, in Lemma C.3 of Jin et al. (2020), they also need to analyze the self-normalized process where the uniform concentration leads to an extra factor of $\sqrt{d}$. The recent work Hu et al. (2022) leverages similar ideas of reference-advantage decomposition, trying to improve the regret bound for the linear MDP but focus on the online setting, whose algorithmic design (optimism v.s. pessimism, choices of weights) and proof techniques (online v.s. offline) are different from ours. Chen et al. (2022) considers the linear mixture MG, which is different from the model considered in this paper. Beyond the linear function approximation, several works on MDP (Jiang et al., 2017; Jin et al., 2021a; Dann et al., 2021; Du et al., 2021; Foster et al., 2021) and MG (Jin et al., 2021b; Huang et al., 2021; Xiong et al., 2022) design algorithms in the online setting with general function approximation. When applied to the linear setting, though their regret bounds are sharper than that of Jin et al. (2020); Xie et al. (2020), their algorithms are only information-theoretic and are computationally inefficient.

**Variance-weighted Regression.** It is known that variance information is essential for sharp horizon dependence (Azar et al., 2017; Zhang et al., 2020; 2021; Zhou et al., 2021). Particularly, for online linear mixture MDP, Zhou et al. (2021) develops the variance-weighted regression and achieves the minimax optimal regret bound; Zhang et al. (2021) considers the time-homogeneous setting and achieves a horizon-free guarantee. This innovative idea is first introduced to the offline setting by Min et al. (2021) and Yin et al. (2022). In this paper, we mainly generalize this technique to the more challenging offline setting without the additional independence assumption required in the existing approaches. See Section 5 for details.

# B  RESULTS FOR MARKOV GAME

We extend our techniques to the linear Markov games in this section.

## B.1  PROBLEM SETUP

We introduce the two-player zero-sum Markov games (MGs) with notations similar to single-agent MDPs, which is a slight abuse of notation but should be clear from the context.

**Two-player Zero-sum Markov Game with Linear Function Approximation** is defined by a tuple $(\mathcal{S}, \mathcal{A}, \mathcal{B}, H, \mathbb{P}, r)$, where $\mathcal{S}$ denotes the state space, $\mathcal{A}$ and $\mathcal{B}$ are the action spaces for the two players, $H$ is the length of each episode, $\mathbb{P} = \{\mathbb{P}_h : \mathcal{S} \times \mathcal{A} \times \mathcal{B} \to \Delta_{\mathcal{S}}\}_{h=1}^{H}$ is the transition kernel, and $r = \{r_h : \mathcal{S} \times \mathcal{A} \times \mathcal{B} \to [0,1]\}_{h=1}^{H}$ is the reward function. The first player (referred to as the max-player) takes action from $\mathcal{A}$ aiming to maximize the cumulative reward, while the second player (referred to as the min-player) want to minimize it. The policy of the max-player is defined as $\pi = \{\pi_h : \mathcal{S} \to \Delta_{\mathcal{A}}\}_{h=1}^{H}$. Analogously, the policy for the min-player is defined by $\nu = \{\nu_h : \mathcal{S} \to \Delta_{\mathcal{B}}\}$.

**Value Function and Nash Equilibrium**. For any fixed policy pair $(\pi, \nu)$, we define the value function $V_h^{\pi,\nu}$ and the Q-function $Q_h^{\pi,\nu}$ as

$$V_h^{\pi,\nu}(x) = \mathbb{E}_{\pi,\nu}[\sum_{h'=h}^{H} r_{h'} | x_h = x], \qquad Q_h^{\pi,\nu}(x,a,b) = \mathbb{E}_{\pi,\nu}[\sum_{h'=h}^{H} r_{h'} | (x_h, a_h, b_h) = (x,a,b)].$$

For any function $V : \mathcal{S} \to \mathbb{R}$, we also define the shorthand notations for the conditional mean, conditional variance, and Bellman operator as follows:

$$(\mathbb{P}_h V)(x,a,b) := \sum_{x' \in \mathcal{S}} \mathbb{P}_h(x'|x,a,b)V(x'), \quad [\mathrm{Var}_h V](x,a,b) := [\mathbb{P}_h V^2](x,a,b) - ([\mathbb{P}_h V](x,a,b))^2,$$

$$(\mathcal{T}_h V)(x,a,b) := r_h(x,a,b) + (\mathbb{P}_h V)(x,a,b).$$

For any max-player's policy $\pi$, we define the best-response as $\mathrm{br}(\pi) = \mathrm{argmin}_\nu V_h^{\pi,\nu}(x)$ for all $(x,h) \in \mathcal{S} \times [H]$. Similarly, we can define $\mathrm{br}(\nu)$ by $\mathrm{br}(\nu) = \mathrm{argmax}_\pi V_h^{\pi,\nu}(x)$ for all $(x,h) \in \mathcal{S} \times [H]$. We say $(\pi^*, \nu^*)$ is a Nash equilibrium (NE) if $\pi^*$ and $\nu^*$ are the best response to each other. For simplicity, we denote $V_h^{\pi,*} = V_h^{\pi,\mathrm{br}(\pi)}$, $V_h^{*,\nu} = V_h^{\mathrm{br}(\nu),\nu}$, and $V_h^*(x) = V_h^{\pi^*,\nu^*}(x)$. It is well known that $(\pi^*, \nu^*)$ is the solution to $\max_\pi \min_\nu V_h^{\pi,\nu}(x)$. Then we can measure the optimality of a policy pair $(\pi, \nu)$ by the duality gap, which is defined in Zhong et al. (2022); Xie et al. (2020) as follows:

$$\mathrm{Gap}((\pi,\nu), x) = V_1^{*,\nu}(x) - V_1^{\pi,*}(x). \tag{11}$$

We consider the linear MG (Xie et al., 2020), which generalizes the definition of linear MDP.

**Definition 2** (Linear MG). *$MG(\mathcal{S}, \mathcal{A}, \mathcal{B}, H, \mathbb{P}, r)$ is a linear MG with a (known) feature map $\phi : \mathcal{S} \times \mathcal{A} \times \mathcal{B} \to \mathbb{R}^d$, if for any $h \in [H]$, there exist $d$ unknown signed measures $\mu_h = \left(\mu_h^{(1)}, \cdots, \mu_h^{(d)}\right)$ over $\mathcal{S}$ and an unknown vector $\theta_h \in \mathbb{R}^d$, such that for any $(x,a) \in \mathcal{S} \times \mathcal{A}$, we have*

$$\mathbb{P}_h(\cdot \mid x,a,b) = \langle \phi(x,a,b), \mu_h(\cdot) \rangle, \quad r_h(x,a,b) = \langle \phi(x,a,b), \theta_h \rangle. \tag{12}$$

*We assume that $\|\phi(x,a,b)\| \le 1$ for all $(x,a,b) \in \mathcal{S} \times \mathcal{A} \times \mathcal{B}$, and $\max\{\|\mu_h(\mathcal{S})\|, \|\theta_h\|\} \le \sqrt{d}$ for all $h \in [H]$.*

With a slight abuse of notation, we make the following coverage assumption for MGs.

**Assumption 3.** *We assume $\kappa = \min_{h \in [H]} \lambda_{\min}(\mathbb{E}_{d_h^b}[\phi(x,a,b)\phi(x,a,b)^\top]) > 0$ for MGs.*

### B.2 LinPMVI-ADV+

LinPMVI-ADV+ (Algorithm 3) is a variant of pessimistic minimax value iteration (PMVI) from (Zhong et al., 2022). At a high level, LinPMVI-ADV+ constructs *pessimistic* estimations of the Q-functions for both players and outputs a policy pair by solving two Nash Equilibrium (NE) based on these two estimated value functions. For linear MG, these can also be done by regressions. Suppose we have constructed value functions $(\underline{V}_{h+1}, \overline{V}_{h+1})$ at $(h+1)$-th step, and two *independent* variance estimators $\underline{\sigma}_h^2$ and $\overline{\sigma}_h^2$, which are constructed similarly to Section 5. For now, let us focus on the main components of Algorithm 3 and defer the construction of the variance estimators to next subsection.

Given (9), we approximate the Bellman equations $\mathcal{T}_h \underline{V}_{h+1}$ and $\mathcal{T}_h \overline{V}_{h+1}$ by solving the following regression problems:

$$
\begin{aligned}
\underline{w}_h &\leftarrow \underset{w \in \mathbb{R}^d}{\mathrm{argmin}} \sum_{\tau \in \mathcal{D}} \frac{[r_h^\tau + \underline{V}_{h+1}(x_{h+1}^\tau) - (\phi_h^\tau)^\top w]^2}{\underline{\sigma}_h^2(x_h^\tau, a_h^\tau, b_h^\tau)} + \lambda \|w\|_2^2, \quad \text{and} \quad \underline{\mathcal{T}}_h \underline{V}_{h+1}(o) := \phi(o)^\top \underline{w}_h, \\
\overline{w}_h &\leftarrow \underset{w \in \mathbb{R}^d}{\mathrm{argmin}} \sum_{\tau \in \mathcal{D}} \frac{[r_h^\tau + \overline{V}_{h+1}(x_{h+1}^\tau) - (\phi_h^\tau)^\top w]^2}{\overline{\sigma}_h^2(x_h^\tau, a_h^\tau, b_h^\tau)} + \lambda \|w\|_2^2, \quad \text{and} \quad \overline{\mathcal{T}}_h \overline{V}_{h+1}(o) := \phi(o)^\top \overline{w}_h
\end{aligned}
\tag{13}
$$

---

**Algorithm 3** LinPMVI-ADV+

1: **Initialize:** Input datasets $\mathcal{D}, \mathcal{D}'$, and $\beta_3$; $\overline{V}_{H+1}(\cdot) = \underline{V}_{H+1}(\cdot) = 0$.
2: Construct variance estimator $\overline{\sigma}_h^2$ and $\underline{\sigma}_h^2$ as in Appendix B.3 with $\mathcal{D}'$.
3: **for** $h = H, \dots, 1$ **do**
4:     $\underline{\Sigma}_h = \sum_{\tau \in \mathcal{D}} \frac{\phi_h^\tau (\phi_h^\tau)^\top}{\underline{\sigma}_h^2(x_h^\tau, a_h^\tau, b_h^\tau)} + \lambda I_d, \overline{\Sigma}_h = \sum_{\tau \in \mathcal{D}} \frac{\phi_h^\tau (\phi_h^\tau)^\top}{\overline{\sigma}_h^2(x_h^\tau, a_h^\tau, b_h^\tau)} + \lambda I_d;$
5:     $\underline{w}_h = \underline{\Sigma}_h^{-1} (\sum_{\tau \in \mathcal{D}} \phi_h^\tau (r_h^\tau + \underline{V}_{h+1}(x_{h+1}^\tau))), \overline{w}_h = \overline{\Sigma}_h^{-1} (\sum_{\tau \in \mathcal{D}} \phi_h^\tau (r_h^\tau + \overline{V}_{h+1}(x_{h+1}^\tau)));$
6:     $\underline{\Gamma}_h(\cdot, \cdot, \cdot) \leftarrow \beta_3 \|\phi(\cdot, \cdot, \cdot)\|_{\underline{\Sigma}_h^{-1}}, \overline{\Gamma}_h(\cdot, \cdot, \cdot) \leftarrow \beta_3 \|\phi(\cdot, \cdot, \cdot)\|_{\overline{\Sigma}_h^{-1}}$
7:     $\underline{Q}_h(\cdot, \cdot, \cdot) \leftarrow \{\phi(\cdot, \cdot, \cdot)^\top \underline{w}_h - \underline{\Gamma}_h(\cdot, \cdot, \cdot)\}_{[0, H-h+1]}.$
8:     $\overline{Q}_h(\cdot, \cdot, \cdot) \leftarrow \{\phi(\cdot, \cdot, \cdot)^\top \overline{w}_h + \overline{\Gamma}_h(\cdot, \cdot, \cdot)\}_{[0, H-h+1]}.$
9:     $(\widehat{\pi}_h(\cdot \mid \cdot), \nu_h'(\cdot \mid \cdot)) \leftarrow \mathrm{NE}(\underline{Q}_h(\cdot, \cdot, \cdot)); \underline{V}_h(\cdot) \leftarrow \langle \underline{Q}_h(\cdot, \cdot, \cdot), \widehat{\pi}_h(\cdot \mid \cdot) \times \nu_h'(\cdot \mid \cdot) \rangle_{\mathcal{A} \times \mathcal{B}}.$
10:    $(\pi_h'(\cdot \mid \cdot), \widehat{\nu}_h(\cdot \mid \cdot)) \leftarrow \mathrm{NE}(\overline{Q}_h(\cdot, \cdot, \cdot)); \overline{V}_h(\cdot) \leftarrow \langle \overline{Q}_h(\cdot, \cdot, \cdot), \pi_h'(\cdot \mid \cdot) \times \widehat{\nu}_h(\cdot \mid \cdot) \rangle_{\mathcal{A} \times \mathcal{B}}.$
11: **end for**
12: **Output:** $(\widehat{\pi} = \{\widehat{\pi}_h\}_{h=1}^H, \widehat{\nu} = \{\widehat{\nu}_h\}_{h=1}^H).$

---

where $(o)$ is short for $(x, a, b)$, and $\overline{\mathbb{P}}_h g_{h+1}, \underline{\mathbb{P}}_h g_{h+1}$ can be obtained by setting $r_h^\tau = 0$ in $\overline{\mathcal{T}}_h g_{h+1}$ and $\underline{\mathcal{T}} g_{h+1}$. Denoting the covariance estimators as $\underline{\Sigma}_h = \sum_{\tau \in \mathcal{D}} \frac{\phi_h^\tau (\phi_h^\tau)^\top}{\underline{\sigma}_h^2(x_h^\tau, a_h^\tau, b_h^\tau)} + \lambda I_d$, and $\overline{\Sigma}_h = \sum_{\tau \in \mathcal{D}} \frac{\phi_h^\tau (\phi_h^\tau)^\top}{\overline{\sigma}_h^2(x_h^\tau, a_h^\tau, b_h^\tau)} + \lambda I_d$, we can estimate the Q-functions by LCB for the max-player and UCB for the min-player, respectively:

$$\underline{Q}_h(o) \leftarrow \phi(o)^\top \underline{w}_h - \underline{\Gamma}_h(x, a, b), \qquad \overline{Q}_h(o) \leftarrow \phi(o)^\top \overline{w}_h + \overline{\Gamma}_h(x, a, b). \tag{14}$$

where we remark that they are pessimistic for the max-player and the min-player, respectively. Next, we solve the matrix games with payoffs $\underline{Q}_h$ and $\overline{Q}_h$:

$$(\widehat{\pi}_h(\cdot \mid \cdot), \nu_h'(\cdot \mid \cdot)) \leftarrow \mathrm{NE}(\underline{Q}_h(\cdot, \cdot, \cdot)), \qquad \text{and} \qquad (\pi_h'(\cdot \mid \cdot), \widehat{\nu}_h(\cdot \mid \cdot)) \leftarrow \mathrm{NE}(\overline{Q}_h(\cdot, \cdot, \cdot)). \tag{15}$$

The V-functions estimations $\underline{V}_h$ and $\overline{V}_h$ are then given by

$$\underline{V}_h = \mathbb{E}_{a \sim \widehat{\pi}_h(\cdot \mid \cdot), b \sim \nu_h'(\cdot \mid \cdot)} \underline{Q}_h(\cdot, a, b) \qquad \text{and} \qquad \overline{V}_h = \mathbb{E}_{a \sim \pi_h'(\cdot \mid \cdot), b \sim \widehat{\nu}_h(\cdot \mid \cdot)} \overline{Q}_h(\cdot, a, b). \tag{16}$$

After $H$ steps, the algorithm outputs the policy pair $(\widehat{\pi} = \{\widehat{\pi}_h\}_{h=1}^H, \widehat{\nu} = \{\widehat{\nu}_h\}_{h=1}^H)$ and value functions $(\underline{V} = \{\underline{V}_h\}_{h=1}^H, \overline{V} = \{\overline{V}_h\}_{h=1}^H)$.

Similar to the linear MDP, a sharper uncertainty bonus leads to a smaller suboptimality gap. The techniques developed for MDPs can be separately applied to the max-player and min-player. Specifically, given $(\underline{V}_{h+1}, \overline{V}_{h+1})$, if we denote the Nash Value as $V_{h+1}^*$, we can decompose the uncertainties as follows:

$$\mathcal{T}_h \underline{V}_{h+1}(o) - \underline{\mathcal{T}}_h \underline{V}_{h+1}(o) = \mathcal{T}_h V_{h+1}^*(o) - \underline{\mathcal{T}}_h V_{h+1}^*(o) + \mathbb{P}_h(\underline{V}_{h+1} - V_{h+1}^*)(o) - \underline{\mathbb{P}}_h(\underline{V}_{h+1} - V_{h+1}^*)(o),$$

$$\mathcal{T}_h \overline{V}_{h+1}(o) - \overline{\mathcal{T}}_h \overline{V}_{h+1}(o) = \underbrace{\mathcal{T}_h V_{h+1}^*(o) - \overline{\mathcal{T}}_h V_{h+1}^*(o)}_{\text{Reference Part}} + \mathbb{P}_h(\overline{V}_{h+1} - V_{h+1}^*)(o) - \overline{\mathbb{P}}_h(\overline{V}_{h+1} - V_{h+1}^*)(o).$$

Then, similar to the single-agent MDP, for sufficiently large $K$, the uncertainty is dominated by the reference part with $V_{h+1}^*$. Moreover, when the independent variance estimators could approximate the conditional variance well, we can set $\overline{\Gamma}_h(x, a, b) = \tilde{O}(\sqrt{d}) \|\phi(x, a, b)\|_{\overline{\Sigma}_h^{-1}}$ and $\underline{\Gamma}_h(x, a, b) = \tilde{O}(\sqrt{d}) \|\phi(x, a, b)\|_{\underline{\Sigma}_h^{-1}}$. The full pseudo code is presented in Algorithm 3 and we have the following theoretical guarantee.

**Theorem 5** (LinPMVI-ADV+). *Under Assumption 1, for $K \geq \tilde{\Omega}(d^2 H^6 / \kappa)$, if we set $0 < \lambda < \kappa$ and $\beta_3 = \tilde{O}(\sqrt{d})$ in Algorithm 3, then with probability at least $1 - \delta$, we have*

$$V_1^{*, \widehat{\nu}}(x) - V_1^{\widehat{\pi}, *}(x) \leq \tilde{O}(\sqrt{d}) \cdot \left( \max_\nu \sum_{h=1}^H \mathbb{E}_{\pi^*, \nu} \|\phi(x_h, a_h, b_h)\|_{\Sigma_h^{*-1}} + \max_\pi \sum_{h=1}^H \mathbb{E}_{\pi, \nu^*} \|\phi(x_h, a_h, b_h)\|_{\Sigma_h^{*-1}} \right),$$

*where $(\pi^*, \nu^*)$ is an NE and $\Sigma_h^* = \sum_{\tau \in \mathcal{D}} \phi_h^\tau (\phi_h^\tau)^\top / [\mathbb{V}_h V_{h+1}^*](x_h^\tau, a_h^\tau, b_h^\tau) + \lambda I_d$.*

---

**Algorithm 4** PMVI (LinPMVI-ADV)

---

1: **Initialize:** Input dataset $\mathcal{D}$, $\beta_3$; $\overline{V}_{H+1}(\cdot) = \underline{V}_{H+1}(\cdot) = 0$.
2: **for** $h = H, \dots, 1$ **do**
3: $\quad \Lambda_h \leftarrow \sum_{\tau \in \mathcal{D}} \phi(x_h^\tau, a_h^\tau, b_h^\tau) \phi(x_h^\tau, a_h^\tau, b_h^\tau)^\top + \lambda I$.
4: $\quad \underline{w}_h \leftarrow \Lambda_h^{-1}(\sum_{\tau \in \mathcal{D}} \phi\,(x_h^\tau, a_h^\tau, b_h^\tau)\,(r_h^\tau + \underline{V}_{h+1}(x_{h+1}^\tau)))$.
5: $\quad \overline{w}_h \leftarrow \Lambda_h^{-1}(\sum_{\tau \in \mathcal{D}} \phi\,(x_h^\tau, a_h^\tau, b_h^\tau)\,(r_h^\tau + \overline{V}_{h+1}(x_{h+1}^\tau)))$.
6: $\quad \Gamma_h(\cdot, \cdot, \cdot) \leftarrow \beta_3 \cdot (\phi(\cdot, \cdot, \cdot)^\top \Lambda_h^{-1} \phi(\cdot, \cdot, \cdot))^{1/2}$.
7: $\quad \underline{Q}_h(\cdot, \cdot, \cdot) \leftarrow \{\phi(\cdot, \cdot, \cdot)^\top \underline{w}_h - \Gamma_h(\cdot, \cdot, \cdot)\}_{[0, H-h+1]}$.
8: $\quad \overline{Q}_h(\cdot, \cdot, \cdot) \leftarrow \{\phi(\cdot, \cdot, \cdot)^\top \overline{w}_h + \Gamma_h(\cdot, \cdot, \cdot)\}_{[0, H-h+1]}$.
9: $\quad (\widehat{\pi}_h(\cdot \mid \cdot), \nu_h'(\cdot \mid \cdot)) \leftarrow \mathrm{NE}(\underline{Q}_h(\cdot, \cdot, \cdot))$.
10: $\quad (\pi_h'(\cdot \mid \cdot), \widehat{\nu}_h(\cdot \mid \cdot)) \leftarrow \mathrm{NE}(\overline{Q}_h(\cdot, \cdot, \cdot))$.
11: $\quad \underline{V}_h(\cdot) \leftarrow \langle \underline{Q}_h(\cdot, \cdot, \cdot), \widehat{\pi}_h(\cdot \mid \cdot) \times \nu_h'(\cdot \mid \cdot)\rangle_{\mathcal{A} \times \mathcal{B}}$.
12: $\quad \overline{V}_h(\cdot) \leftarrow \langle \overline{Q}_h(\cdot, \cdot, \cdot), \pi_h'(\cdot \mid \cdot) \times \widehat{\nu}_h(\cdot \mid \cdot)\rangle_{\mathcal{A} \times \mathcal{B}}$.
13: **end for**
14: **Output:** $(\widehat{\pi} = \{\widehat{\pi}_h\}_{h=1}^H, \widehat{\nu} = \{\widehat{\nu}_h\}_{h=1}^H)$.

---

Similar with the single-agent MDPs, LinPMVI-ADV+ replace an explicit dependence on the planning horizon $H$ in PMVI (Zhong et al., 2022) with an instance-dependent characterization through $\Sigma_h^*$. The instance-dependent bound of LinPMVI-ADV+ is never worse than that of PMVI, and the improvement is strict when specialized to the special case of tabular setting.

**A tighter lower bound.** To further interpret the result, we establish the following nearly matching lower bound which tightens that in Zhong et al. (2022).

**Theorem 6** (Lower Bound for MG). *Fix horizon $H$, dimension $d$, probability $\delta > 0$, and sample size $K \geq \tilde{\Omega}(d^4)$. There exists a class $\mathcal{M}$ of linear MGs and an offline dataset $\mathcal{D}$ with $|\mathcal{D}| = K$, such that for any policy pair $(\widehat{\pi}, \widehat{\nu})$, it holds with probability at least $1 - \delta$ that*

$$\sup_{M \in \mathcal{M}} \mathbb{E}_M[V_1^{*, \widehat{\nu}}(x_1) - V_1^{\widehat{\pi}, *}(x_1)] \geq c\sqrt{d} \cdot \Big( \max_\nu \sum_{h=1}^H \mathbb{E}_{\pi^*, \nu} \|\phi_h\|_{\Sigma_h^{*-1}} + \max_\pi \sum_{h=1}^H \mathbb{E}_{\pi, \nu^*} \|\phi_h\|_{\Sigma_h^{*-1}} \Big),$$

*where $\Sigma_h^* = \sum_{\tau \in \mathcal{D}} \phi_h^\tau (\phi_h^\tau)^\top / [\mathbb{V}_h V_{h+1}^*](x_h^\tau, a_h^\tau, b_h^\tau) + \lambda I_d$ and $c > 0$ is a universal constant.*

As $[\mathbb{V}_h V_{h+1}^*](\cdot, \cdot, \cdot) \geq 1$, it holds that $\|\phi(x_h, a_h, b_h)\|_{\Sigma_h^{*-1}} \geq \|\phi(x_h, a_h, b_h)\|_{\Lambda_h^{-1}}$. Therefore, Theorem 6 improves the lower bound in Zhong et al. (2022) at least by a factor of $\sqrt{d}$. Moreover, LinPMVI-ADV+ matches this lower bound up to logarithmic factor thus is nearly minimax optimal when $K$ exceeds the threshold specified in the theorem.

### B.3 LINPMVI-ADV

In this subsection, we first present the full pseudo code of PMVI (Zhong et al., 2022). We first follow the reasoning in last subsection but with naive variance 1, and thus with the regular ridge regression. Then, the reference-advantage decomposition allows us to improve PMVI by a factor of $\sqrt{d}$ by invoking Lemma 9 without a uniform concentration in the reference part. The resulting bonus is $\Gamma_h(\cdot, \cdot, \cdot) = \tilde{O}(\sqrt{d}H) \|\phi(\cdot, \cdot, \cdot)\|_{\Lambda_h^{-1}}$.

LinPMVI-ADV admits the following theoretical guarantee:

**Theorem 7** (LinPMVI-ADV). *Under Assumption 1, for $K > \tilde{\Omega}(d^2 H^2 / \kappa)$, if we set $\lambda = 1$ and $\beta_4 = \tilde{O}(\sqrt{d}H)$ in Algorithm 4, then with probability at least $1 - \delta$, we have*

$$V_1^{*, \widehat{\nu}}(x) - V_1^{\widehat{\pi}, *}(x) \leq \tilde{O}(\sqrt{d}H) \cdot \Big( \max_\nu \sum_{h=1}^H \mathbb{E}_{\pi^*, \nu} \|\phi_h\|_{\Lambda_h^{-1}} + \max_\pi \sum_{h=1}^H \mathbb{E}_{\pi, \nu^*} \|\phi_h\|_{\Lambda_h^{-1}} \Big), \quad (17)$$

*where $(\pi^*, \nu^*)$ is an NE and $\Lambda_h = \sum_{\tau \in \mathcal{D}} \phi(x_h^\tau, a_h^\tau, b_h^\tau) \phi(x_h^\tau, a_h^\tau, b_h^\tau)^\top + \lambda I_d$.*

We now proceed to construct the variance estimator for Algorithm 3.

**Construction of the Variance Estimators.** To begin with, we run Algorithm 4 to construct $\{\overline{V}'_h, \underline{V}'_h\}_{h=1}^H$ with an independent dataset $\mathcal{D}'$. This will only incur a factor of 2 in the final sample complexity. Similar to Lemma 1, we can show that there exist $\beta_{h,1}(\overline{V}'_{h+1})$ and $\beta_{h,2}(\overline{V}'_{h+1})$ such that $[\mathbb{P}_h(\overline{V}'_{h+1})^2](x,a) = \left\langle \phi(x,a), \beta_{h,2}(\overline{V}'_{h+1}) \right\rangle$ and $[\mathbb{P}_h \overline{V}'_{h+1}](x,a) = \left\langle \phi(x,a), \beta_{h,1}(\overline{V}'_{h+1}) \right\rangle$. We approximate them via ridge regression with $\mathcal{D}'$:

$$\widetilde{\beta}_{h,2} = \underset{\beta \in \mathbb{R}^d}{\operatorname{argmin}} \sum_{\tau \in \mathcal{D}'} \left[ \langle \phi_h^\tau, \beta \rangle - (\overline{V}'_{h+1})^2 \left( x_{h+1}^\tau \right) \right]^2 + \lambda \|\beta\|_2^2,$$

$$\widetilde{\beta}_{h,1} = \underset{\beta \in \mathbb{R}^d}{\operatorname{argmin}} \sum_{\tau \in \mathcal{D}'} \left[ \langle \phi_h^\tau, \beta \rangle - \overline{V}'_{h+1} \left( x_{h+1}^\tau \right) \right]^2 + \lambda \|\beta\|_2^2.$$

Then, the variance estimator is then constructed as

$$\overline{\sigma}_h^2(x,a) := \max \left\{ 1, \left[ \phi(x,a)^\top \widetilde{\beta}_{h,2} \right]_{[0,H^2]} - \left[ \phi(x,a)^\top, \widetilde{\beta}_{h,1} \right]_{[0,H]}^2 - \tilde{O}\left( \frac{dH^3}{\sqrt{K\kappa}} \right) \right\}.$$

Similarly, we can construct the variance estimator $\underline{\sigma}_h^2(x,a)$ with $\underline{V}'_{h+1}$ and dataset $\mathcal{D}'$. In particular, as $\overline{\sigma}_h(\cdot,\cdot)$ and $\underline{\sigma}_h(\cdot,\cdot)$ only depend on the dataset $\mathcal{D}'$, they are independent of $\mathcal{D}$. The variance estimation error is characterized in Lemma 7 and the proof of MGs is presented in Appendix F.

## C   AUXILIARY LEMMAS

In this section, we provide several useful lemmas to facilitate the proof of linear MDP. The first lemma states that if we adopt the principle of pessimism, the suboptimality bound essentially reduces to the the uncertainty estimation, i.e., construction of the bonuses.

**Lemma 2** (Regret Decomposition Lemma for MDP). *Under the condition that with probability at least $1 - \delta$, the functions $\Gamma_h : \mathcal{S} \times \mathcal{A} \to \mathbb{R}$ in Algorithms 2 and 1 ($\Gamma_h = b_{0,h} + b_{1,h}$) satisfying*

$$|\mathcal{T}_h \widehat{V}_{h+1}(x,a) - \widehat{\mathcal{T}}_h \widehat{V}_{h+1}(x,a)| \le \Gamma_h(x,a), \quad \forall (x,a,h) \in \mathcal{S} \times \mathcal{A} \times [H],$$

*we have that with probability at least $1 - \delta$, for Algorithm 2 and Algorithm 1 that for any $x \in \mathcal{S}$,*

$$V_1^*(x) - V_1^{\widehat{\pi}}(x) \le V_1^*(x) - \widehat{V}_1(x) \le 2 \sum_{h=1}^H \mathbb{E}_{\pi^*}[\Gamma_h(x_h, a_h) \mid x_1 = x].$$

*Proof.* See Lemma 3.1 and Theorem 4.2 in Jin et al. (2021c) for a detailed proof. $\qquad \square$

**Lemma 3** (Decomposition). *For a function $f_{h+1}$, suppose that $\mathcal{T}_h f_{h+1}(\cdot,\cdot) = \phi(\cdot,\cdot)^\top w_h$. With $\widehat{w}_h = \Sigma_h^{-1} \left( \sum_{\tau \in \mathcal{D}} \frac{\phi(x_h^\tau, a_h^\tau) \cdot (r_h^\tau + f_{h+1}(x_{h+1}^\tau))}{\widehat{\sigma}_h^2(x_h^\tau, a_h^\tau)} \right)$ where $\widehat{\sigma}_h^2(x_h^\tau, a_h^\tau)$ can be either 1 (for regular ridge regression) or the estimated variance (for variance-weighted ridge regression) and $\Sigma_h = \sum_{\tau \in \mathcal{D}} \frac{\phi(x_h^\tau, a_h^\tau) \phi(x_h^\tau, a_h^\tau)^\top}{\widehat{\sigma}_h^2(x_h^\tau, a_h^\tau)} + \lambda I_d$. Then, we have the following decomposition:*

$$\left( \mathcal{T}_h f_{h+1} - \widehat{\mathcal{T}}_h f_{h+1} \right)(x,a) = \phi(x,a)^\top w_h - \phi(x,a)^\top \widehat{w}_h$$

$$\le \lambda \|w_h\|_{\Sigma_h^{-1}} \|\phi(x,a)\|_{\Sigma_h^{-1}} + \| \sum_{\tau \in \mathcal{D}} \frac{\phi(x_h^\tau, a_h^\tau)}{\widehat{\sigma}_h(x_h^\tau, a_h^\tau)} \xi_h^\tau(f_{h+1}) \|_{\Sigma_h^{-1}} \|\phi(x,a)\|_{\Sigma_h^{-1}}, \quad (18)$$

*where $\xi_h^\tau(f_{h+1}) = \frac{r_h^\tau + f_{h+1}(x_{h+1}^\tau) - \mathcal{T}_h f_{h+1}(x_h^\tau, a_h^\tau)}{\widehat{\sigma}_h(x_h^\tau, a_h^\tau)}$. In particular, if $|f_{h+1}|$ is bounded by $H - 1$ and $\| \sum_{\tau \in \mathcal{D}} \frac{\phi(x_h^\tau, a_h^\tau)}{\widehat{\sigma}_h(x_h^\tau, a_h^\tau)} \cdot \xi_h^\tau(f_{h+1}) \|_{\Sigma_h^{-1}} \le \beta$, then we can set $\lambda$ sufficiently small so that $\sqrt{\lambda} dH \le \beta$. In this case, the second term is dominating. $\mathbb{P}_h f_{h+1}(\cdot,\cdot)$ admits similar results by setting $r_h \equiv 0$.*

*Proof.* See Appendix I for a detailed proof. $\qquad \square$

## D    PROOFS OF LINPEVI-ADV

### D.1    PROOF OF THEOREM 1

The proof requires a more refined induction analysis to deal with the temporal dependency. For instance, when analyzing the $h$-step, we cannot take the condition on that $\left\|\widehat{V}_{h+1} - V_{h+1}^*\right\|$ is small, which is required to make sure that the uncertainty of the advantage function is non-dominating. This is because due to the temporal dependency, taking condition on $\widehat{V}_{h+1}$ may influence the distribution at step $h$. A carefully crafted induction analysis is employed to solve the challenge.

*Proof of Theorem 1.* We will prove the theorem by induction. For $h = H$, for a function $\|g_{h+1}\|_\infty \leq R - 1$, we invoke Lemma 3:

$$\left|\mathcal{T}_h g_{h+1} - \widehat{\mathcal{T}}_h g_{h+1}\right|(x,a) \leq \underbrace{\sqrt{\lambda d} R \left\|\phi(x,a)\right\|_{\Sigma_h^{-1}}}_{(a)} + \underbrace{\left\|\sum_{\tau \in \mathcal{D}} \phi\left(x_h^\tau, a_h^\tau\right) \cdot \xi_h^\tau(g_{h+1})\right\|_{\Sigma_h^{-1}} \left\|\phi(x,a)\right\|_{\Sigma_h^{-1}}}_{(b)}.$$

For the reference part with $g_{h+1} = V_{H+1}^*$, since $V_{H+1}^*$ is independent of $\mathcal{D}$, we can directly apply Lemma 9 with $\lambda = 1$ to obtain that with probability at least $1 - \delta_H = 1 - \delta/H^2$,

$$\left|\mathcal{T}_H V_{H+1}^* - \widehat{\mathcal{T}}_H V_{H+1}^*\right|(x,a) \leq 2\sqrt{d} H \sqrt{\iota} \left\|\phi(x,a)\right\|_{\Lambda_H^{-1}},$$

where $\iota = \log(2H^2 K/\delta) \geq 1$. To simplify the proof, we set $b_{0,H}(x,a) = 3\sqrt{d} H \sqrt{\iota} \left\|\phi(x,a)\right\|_{\Lambda_H^{-1}}$ to further capture the uncertainty of (a) from the advantage function. Then, we can focus on the analysis of (b) for the advantage function. By construction, we have $\mathcal{E}_{H+1} = \{0 \leq V_{H+1}^* - \widehat{V}_{H+1} \leq \frac{8\sqrt{d} H \cdot 0 \sqrt{\iota}}{\sqrt{K\kappa}}\}$ holds with probability 1. By Lemma 9, it suffices to set $b_{1,H}(x,a) = \frac{8d^{3/2} H^2 \iota}{\sqrt{K\kappa}} \left\|\phi(x,a)\right\|_{\Lambda_H^{-1}}$. It follows that we can set

$$\Gamma_H(\cdot,\cdot) = b_{0,H}(\cdot,\cdot) + b_{1,H}(\cdot,\cdot) = \left(3\sqrt{d} H \sqrt{\iota} + \frac{8d^{3/2} H^2 \iota}{\sqrt{K\kappa}}\right) \left\|\phi(\cdot,\cdot)\right\|_{\Lambda_H^{-1}} \leq 4\sqrt{d} H \sqrt{\iota} \left\|\phi(\cdot,\cdot)\right\|_{\Lambda_H^{-1}},$$

where we use $K \geq \tilde{\Omega}(d^2 H^2/\kappa)$ to obtain the last inequality. If pessimism at step $H$ is achieved, we know that

$$Q_H^*(x,a) = \mathcal{T}_H V_{H+1}^*(x,a) \geq \mathcal{T}_H \widehat{V}_{H+1}(x,a) \geq \widehat{Q}_H(x,a), \forall(x,a) \in \mathcal{S} \times \mathcal{A},$$

where the last step is due to $|\mathcal{T}_h \widehat{V}_{H+1}(x,a) - \widehat{\mathcal{T}}_h \widehat{V}_{H+1}(x,a)| \leq \Gamma_H(x,a)$. This implies that $V_H^*(x) \geq \widehat{V}_H(x)$ for all $x \in \mathcal{S}$ and we proceed to bound the error as follows:

$$V_H^*(x) - \widehat{V}_H(x) = \left\langle Q_H^*(x,\cdot) - \widehat{Q}_H(x,\cdot), \pi_H^*(\cdot|x)\right\rangle + \left\langle \widehat{Q}_H(x,\cdot), \pi_H^*(\cdot|x) - \widehat{\pi}_H(\cdot|x)\right\rangle$$

$$\leq \left\langle \mathcal{T}_H \widehat{V}_{H+1}(x,\cdot) - \widehat{\mathcal{T}}_H \widehat{V}_{H+1}(x,\cdot) + \Gamma_H(x,\cdot), \pi_H^*(\cdot|x)\right\rangle + \left\langle \mathcal{T}_H(V_{h+1}^* - \widehat{V}_{H+1})(x,\cdot), \pi_H^*(\cdot|x)\right\rangle$$

$$\leq 2\mathbb{E}_{\pi^*}[\Gamma_H(\cdot,\cdot)|x_H = x] + 0,$$

$$\leq \frac{8\sqrt{d} H \cdot 1\sqrt{\iota}}{\sqrt{K\kappa}} := R_H, \forall x \in \mathcal{S},$$

where the last inequality uses Lemma 13. To summarize, the event $\mathcal{E}_H = \{0 \leq V_H^*(x) - \widehat{V}_H(x) \leq R_H, \forall x \in \mathcal{S}\}$ holds with probability at least $1 - \delta_H = 1 - \frac{\delta}{H^2}$. This is the base case. Now suppose that the event $\mathcal{E}_{h+1} = \{0 \leq V_{h+1}^*(x) - \widehat{V}_{h+1}(x) \leq R_{h+1} := \frac{8\sqrt{d} H(H-h)\sqrt{\iota}}{\sqrt{K\kappa}}, \forall x \in \mathcal{S}\}$ holds with probability at least $1 - \delta_{h+1}$. We are going to establish the result for step $h$. Clearly, we can still set $b_{0,h}(\cdot,\cdot) = 3\sqrt{d} H \sqrt{\iota} \left\|\phi(\cdot,\cdot)\right\|_{\Lambda_h^{-1}}$. It remains to determine $b_{1,h}(\cdot,\cdot)$ for (b) of the advantage function and to ensure that it is non-dominating. We need to deal with the temporal dependency, which requires a more involved analysis. We first state the following lemma.

**Lemma 4** (Lemma B.2 of Jin et al. (2021c)). *Let $f : \mathcal{S} \to [0, R-1]$ be any fixed function. For any $\delta \in (0,1)$, we have*

$$\mathbb{P}\Big(\big\| \sum_{\tau \in \mathcal{D}} \phi_h^\tau \cdot \xi_h^\tau(f) \big\|_{\Lambda_h^{-1}}^2 \geq R^2(2\log(\frac{1}{\delta}) + d\log(1 + \frac{K}{\lambda}))\Big) \leq \delta.$$

However, as $(\widehat{V}_{h+1} - V_{h+1}^*)$ is correlated to $\{(x_h^\tau, a_h^\tau, x_{h+1}^\tau)\}_{\tau \in \mathcal{D}}$, we need a uniform concentration argument. In particular, we remark that we cannot take condition that $V_{h+1}^* - \widehat{V}_{h+1} \leq R_{h+1}$ directly. We consider the function class

$$\mathcal{V}_h(D, B, \lambda) = \{V_h(x; \theta, \beta, \Sigma) : \mathcal{S} \to [0, H] \text{ with } \|\theta\| \leq D, \beta \in [0, B], \Sigma \succeq \lambda \cdot I\},$$

$$\text{where } V_h(x; \theta, \beta, \Sigma) = \max_{a \in \mathcal{A}} \Big\{ \phi(x,a)^\top \theta - \beta \cdot \sqrt{\phi(x,a)^\top \Sigma^{-1} \phi(x,a)} \Big\}_{[0,H-h+1]}. \tag{19}$$

With $V_{h+1}^* - \widehat{V}_{h+1}$ denoted as $f$, we can estimate $D$ as follows:

$$\begin{aligned}
\|\widehat{w}_h(f)\| &= \Big\| \Lambda_h^{-1} \Big( \sum_{\tau \in \mathcal{D}} \phi(x_h^\tau, a_h^\tau) \cdot (r_h^\tau + f(x_{h+1}^\tau)) \Big) \Big\| \\
&\leq H \sum_{\tau \in \mathcal{D}} \sqrt{\phi(x_h^\tau, a_h^\tau)^\top \Lambda_h^{-1/2} \Lambda_h^{-1} \Lambda_h^{-1/2} \phi(x_h^\tau, a_h^\tau)} \\
&\leq H \sqrt{\frac{K}{\lambda}} \sqrt{\sum_{\tau \in \mathcal{D}} \phi(x_h^\tau, a_h^\tau)^\top \Lambda_h^{-1} \phi(x_h^\tau, a_h^\tau)} \\
&= H \sqrt{\frac{K}{\lambda}} \sqrt{\text{Tr}\left(\Lambda_h^{-1}(\Lambda_h - \lambda I_d)\right)} \leq H \sqrt{\frac{Kd}{\lambda}}.
\end{aligned}$$

It follows that

$$f = V_{h+1}^* - \widehat{V}_{h+1} \in \mathcal{F}_{h+1} := \{V_{h+1}^* - V_{h+1} : V_{h+1} \in \mathcal{V}_{h+1}(D_0, B_0, \lambda)\}$$

where $D_0 = H\sqrt{\frac{Kd}{\lambda}}, \lambda = 1$, and $B_0 = 8\sqrt{d}H\sqrt{\iota}$. For any $\epsilon > 0$, we denote the $\epsilon$-cover of $\mathcal{F}_{h+1}$ with respect to the supremum norm as $\mathcal{N}_{h+1}(\epsilon)$ (short for $\mathcal{N}_{h+1}(\epsilon; D, B, \lambda)$) and its $\epsilon$-covering number as $|\mathcal{N}_{h+1}(\epsilon)|$. For each $f \in \mathcal{F}_{h+1}$, we can find $f_\epsilon \in \mathcal{N}_{h+1}(\epsilon)$, such that $\sup_{x \in \mathcal{S}} |f(x) - f_\epsilon(x)| \leq \epsilon$. It follows that

$$\begin{aligned}
&\Big\| \sum_{\tau \in \mathcal{D}} \phi_h^\tau \cdot \xi_h^\tau(f) \Big\|_{\Lambda_h^{-1}}^2 \mathbf{1}\{\|f\|_\infty \leq R_{h+1}\} \\
&\leq 2 \Big\| \sum_{\tau \in \mathcal{D}} \phi_h^\tau \cdot \xi_h^\tau(f_\epsilon) \Big\|_{\Lambda_h^{-1}}^2 \mathbf{1}\{\|f_\epsilon\|_\infty \leq (R_{h+1} + \epsilon)\} + 2 \Big\| \sum_{\tau \in \mathcal{D}} \phi_h^\tau \cdot (\xi_h^\tau(f) - \xi_h^\tau(f_\epsilon)) \Big\|_{\Lambda_h^{-1}}^2 \\
&\leq 2 \Big\| \sum_{\tau \in \mathcal{D}} \phi_h^\tau \cdot \xi_h^\tau(f_\epsilon) \Big\|_{\Lambda_h^{-1}}^2 \mathbf{1}\{\|f_\epsilon\|_\infty \leq (R_{h+1} + \epsilon)\} + 2\epsilon^2 K^2/\lambda,
\end{aligned}$$

where the first inequality uses $\{\|f\|_\infty \leq R_{h+1}\}$ implies $\{\|f_\epsilon\|_\infty \leq (R_{h+1} + \epsilon)\}$ and the second inequality is because the following estimation of the second term:

$$\begin{aligned}
2 \Big\| \sum_{\tau \in \mathcal{D}} \phi_h^\tau \cdot (\xi_h^\tau(f) - \xi_h^\tau(f_\epsilon)) \Big\|_{\Lambda_h^{-1}}^2 &\leq 2\epsilon^2 \sum_{\tau, \tau'=1}^K |\phi_h^\tau \Lambda_h^{-1} \phi_h^{\tau'}| \leq 2\epsilon^2 \|\phi_h^\tau\| \|\phi_h^{\tau'}\| \big\| \Lambda_h^{-1} \big\|_{\text{op}} \\
&\leq 2\epsilon^2 K^2/\lambda,
\end{aligned}$$

where $\|\cdot\|_{\mathrm{op}}$ denotes the operator norm and $\left\|\Lambda_h^{-1}\right\|_{\mathrm{op}} \leq \lambda^{-1}$. With a union bound over $\mathcal{N}_{h+1}(\epsilon)$ and Lemma 4, we obtain that

$$
\mathbb{P}\left(\sup_{f_\epsilon \in \mathcal{N}_{h+1}(\epsilon)} \left\|\sum_{\tau \in \mathcal{D}} \phi_h^\tau \cdot \xi_h^\tau(f_\epsilon)\right\|_{\Lambda_h^{-1}}^2 \mathbf{1}\left\{\|f_\epsilon\|_\infty \leq R_{h+1} + \epsilon\right\} > \right.
$$
$$
\left. (R_{h+1} + \epsilon)^2 (2\log(\frac{H^2 \cdot |\mathcal{N}_{h+1}(\epsilon)|}{\delta}) + d\log(1 + \frac{K}{\lambda}))\right) \leq \delta/H^2.
$$

With probability at least $1 - \delta/H^2$, for all $f \in \mathcal{F}_{h+1}$, we have

$$
\left\|\sum_{\tau \in \mathcal{D}} \phi_h^\tau \cdot \xi_h^\tau(f)\right\|_{\Lambda_h^{-1}}^2 \mathbf{1}\left\{\|f\|_\infty \leq R_{h+1}\right\}
$$
$$
\leq \inf_{\epsilon > 0}\left\{\left(\frac{8\sqrt{d}H^2\sqrt{\iota}}{\sqrt{K\kappa}} + \epsilon\right)^2\left(2\log(\frac{H^2 \cdot |\mathcal{N}_{h+1}(\epsilon)|}{\delta}) + d\log(1 + \frac{K}{\lambda})\right) + 2\epsilon^2 K^2/\lambda\right\} \quad (20)
$$
$$
\leq \left(\frac{128dH^4\iota}{K\kappa}\right)\left(2\log(H^2/\delta) + 4d^2\log(\frac{512K^3\iota}{d^{3/2}H^2})\right) + \frac{2d^3H^4}{K\kappa}
$$
$$
\leq \left(\frac{256dH^4\iota}{K\kappa}\right)\left(2\log(H^2/\delta) + 4d^2\log(\frac{512K^3\iota}{d^{3/2}H^2})\right),
$$

where the second inequality is because the covering number of $\mathcal{F}_{h+1}$ is bounded by that of $\mathcal{V}_{h+1}(D, B, \lambda)$ and by Lemma 11, we can take $\epsilon = d^{3/2}H^2/(K^{3/2}\sqrt{\kappa})$ to obtain

$$
\log|\mathcal{N}_{h+1}(\epsilon)| \leq d\log(1 + \frac{4K^2\sqrt{\kappa}}{dH}) + d^2\log(1 + \frac{512K^3\kappa\iota}{d^{3/2}H^2}) \leq 2d^2\log(\frac{512K^3\iota}{d^{3/2}H^2}),
$$

where the second inequality holds when $K > \sqrt{d}H/(128\sqrt{\kappa}\iota)$. As $\iota > 1$ and $\log\iota \leq \iota$, it further holds that

$$
\left(\frac{256dH^4\iota}{K\kappa}\right)\left(2\log(H^2/\delta) + 4d^2\log(\frac{512K^3\iota}{d^{3/2}H^2})\right) \leq \left(\frac{256dH^4\iota}{K\kappa}\right)\left(2\iota + 4d^2\left(\iota + \log(512)\iota + 3\iota\right)\right)
$$
$$
\leq \frac{8704d^3H^4\iota^2}{K\kappa},
$$

To summarize, it suffices to set $b_{1,h} = \frac{94d^{3/2}H^2\iota}{\sqrt{K\kappa}}\|\phi(x, a)\|_{\Lambda_h^{-1}}$ and we have

$$
\mathbb{P}\left(\left\|\sum_{\tau \in \mathcal{D}} \phi_h^\tau \xi_h^\tau(V_{h+1}^* - \widehat{V}_{h+1})\right\| > \frac{94d^{3/2}H^2\iota}{\sqrt{K\kappa}}\right)
$$
$$
\leq \mathbb{P}\left(\left\|\sum_{\tau \in \mathcal{D}} \phi_h^\tau \xi_h^\tau(V_{h+1}^* - \widehat{V}_{h+1})\right\| \mathbf{1}\left\{\left\|V_{h+1}^* - \widehat{V}_{h+1}\right\|_\infty \leq R_{h+1}\right\} > \frac{94d^{3/2}H^2\iota}{\sqrt{K\kappa}}\right)
$$
$$
+ \mathbb{P}\left(\mathbf{1}\left\{\left\|V_{h+1}^* - \widehat{V}_{h+1}\right\|_\infty > R_{h+1}\right\}\right)
$$
$$
\leq \delta/H^2 + \delta_{h+1} := \delta_h,
$$

where $\delta_{h+1}$ is the failure probability at step $h + 1$. As $K > \tilde{\Omega}\left(d^2H^2/\kappa\right)$, we can set

$$
\Gamma_h(\cdot, \cdot) \leq \left(3\sqrt{d}H\sqrt{\iota} + \frac{94d^{3/2}H^2\iota}{\sqrt{K\kappa}}\right)\|\phi(\cdot, \cdot)\|_{\Lambda_h^{-1}} \leq 4\sqrt{d}H\sqrt{\iota}\|\phi(\cdot, \cdot)\|_{\Lambda_h^{-1}},
$$

With $R_h := \frac{8\sqrt{d}H(H-h+1)\sqrt{\iota}}{\sqrt{K\kappa}}$, we proceed to analyze the failure probability of $\mathcal{E}_h = \{0 \leq V_{h+1}^*(x) - \widehat{V}_{h+1}(x) \leq R_h, \forall x \in \mathcal{S}\}$. First of all, if $|\mathcal{T}_h\widehat{V}_{h+1} - \widehat{\mathcal{T}}_h\widehat{V}_{h+1}| \leq \Gamma_h$ and event $\mathcal{E}_{h+1}$ holds, we know that

$$
Q_h^*(x, a) = \mathcal{T}_h V_{h+1}^*(x, a) \geq \mathcal{T}_h\widehat{V}_{h+1}(x, a) \geq \widehat{Q}_h(x, a), \forall(x, a) \in \mathcal{S} \times \mathcal{A},
$$

and thus $V_h^*(x) \geq \widehat{V}_h(x)$ for all $x \in \mathcal{S}$. We also have

$$
\begin{aligned}
V_h^*(x) - \widehat{V}_h(x) &= \left\langle Q_h^*(x, \cdot) - \widehat{Q}_h(x, \cdot), \pi_h^*(\cdot|x) \right\rangle + \left\langle \widehat{Q}_h(x, \cdot), \pi_h^*(\cdot|x) - \widehat{\pi}_h(\cdot|x) \right\rangle \\
&\leq \left\langle \mathcal{T}_h \widehat{V}_{h+1}(x, \cdot) - \widehat{\mathcal{T}}_h \widehat{V}_{h+1}(x, \cdot) + \Gamma_h(x, \cdot), \pi_h^*(\cdot|x) \right\rangle + \left\langle \mathcal{T}_h(V_{h+1}^* - \widehat{V}_{h+1})(x, \cdot), \pi_h^*(\cdot|x) \right\rangle \\
&\leq 2\mathbb{E}_{\pi^*}[\Gamma_h(\cdot, \cdot)|x_h = x] + R_{h+1}, \\
&\leq \frac{8\sqrt{d}H \cdot (H - h)\sqrt{\iota}}{\sqrt{K\kappa}} := R_h, \forall x \in \mathcal{S}.
\end{aligned}
$$

Therefore, the failure probability at step $h$ can be upper bounded as

$$
\mathbb{P}\left(\mathcal{E}_h^c\right) \leq \mathbb{P}\left(\mathcal{E}_{h+1}^c \cup \mathcal{E}_{h+1} \cap \{\Gamma_h(\cdot, \cdot) \text{ does not ensures pessimism}\}\right) \leq \delta_{h+1} + \frac{\delta}{H^2} = \delta_h.
$$

Therefore, we have shown that with probability at least $1 - \delta_h = 1 - \delta_{h+1} - \frac{\delta}{H^2}$, pessimism is achieved at step $h$ and $\mathcal{E}_h$ holds with probability at least $1 - \delta_h$. By induction, and a union bound over $h \in [H]$, we know that if we set $\Gamma_h = 4\sqrt{d}H\sqrt{\iota} \|\phi(\cdot, \cdot)\|_{\Lambda_h^{-1}}$, then with probability at least $1 - (\delta_H + \cdots + \delta_1) = 1 - \frac{\delta H(H+1)}{2H^2} > 1 - \delta$, $|\mathcal{T}_h \widehat{V}_{h+1} - \widehat{\mathcal{T}}_h \widehat{V}_{h+1}|(x, a) \leq \Gamma_h(x, a)$ holds for all $(h, x, a) \in [H] \times \mathcal{S} \times \mathcal{A}$. The theorem then follows from Lemma 2. $\qquad\square$

### D.2 PROOF OF THEOREM 4

*Proof of Theorem 4.* The proof basically follows the same arguments of that of Theorem 1 except that we can leverage the finite feature condition to derive a different bonus term for the dominating reference function. We focus on deriving the $\Gamma_h(\cdot, \cdot)$ and omit other details for simplicity.

We first elaborate Eqn. (6.1) via the proof of Lemma 3 (see Section I for details):

$$
\left|\mathcal{T}_h V_{h+1}^* - \widehat{\mathcal{T}}_h V_{h+1}^*\right|(x, a) \leq \underbrace{\sqrt{\lambda d}H \|\phi(x, a)\|_{\Sigma_h^{-1}}}_{(a)} + \underbrace{\sum_{\tau \in \mathcal{D}} \left\langle \phi(x, a), \Lambda_h^{-1}\phi_h^\tau \right\rangle \xi_h^\tau(V_{h+1}^*)}_{(b)},
$$

where $\xi_h^\tau(V_{h+1}^*) = r_h^\tau + V_{h+1}^*(x_{h+1}^\tau) - \mathcal{T}_h V_{h+1}^*(x_h^\tau, a_h^\tau)$. We will bound (b) by $\beta \|\phi(x, a)\|_{\Lambda_h^{-1}}$ with some $\beta > 0$ and we can set $\lambda$ sufficiently small so that $\sqrt{\lambda d}H \leq \beta$. Therefore, we can focus on the analysis of (b).

We denote the state-action pairs at step $h$ as $\mathcal{D}_h = \{(x_h^\tau, a_h^\tau)\}_{\tau \in \mathcal{D}}$. The key is that because $V_{h+1}^*$ is deterministic, conditioned on $\mathcal{D}_h$, the only randomness is from $x_{h+1}^\tau$ and $\{\xi_h^\tau(V_{h+1}^*)\}_{\tau \in \mathcal{D}}$ are still independent and bounded random variables. In particular, for any fixed $\mathcal{D}_h = D_h$ and fixed $\phi(x, a)$, by the Hoeffding's inequality, with probability at least $1 - \frac{\delta}{HM}$, we have

$$
\begin{aligned}
(b) &\leq \sqrt{\sum_{\tau \in \mathcal{D}} H^2 \left\langle \phi(x, a), \Lambda_h^{-1}\phi(x_h^\tau, a_h^\tau) \right\rangle^2 \log\left(\frac{2H^2M}{\delta}\right)} \\
&= H\sqrt{\left(\|\phi(x, a)\|_{\Lambda_h^{-1}}^2 - \lambda \|\phi(x, a)\|_{\Lambda_h^{-2}}^2\right) \log\left(\frac{2H^2M}{\delta}\right)} \\
&\leq H\sqrt{\log\left(\frac{2H^2M}{\delta}\right)} \|\phi(x, a)\|_{\Lambda_h^{-1}}, \qquad \forall (x, a, h) \in \mathcal{S} \times \mathcal{A} \times [H],
\end{aligned}
$$

where in the equality, we use

$$
\begin{aligned}
\sum_{\tau \in \mathcal{D}} \left\langle \phi(x, a), \Lambda_h^{-1}\phi(x_h^\tau, a_h^\tau) \right\rangle^2 &= \sum_{\tau \in \mathcal{D}} \phi(x, a)^\top \Lambda_h^{-1}\phi(x_h^\tau, a_h^\tau)\phi(x_h^\tau, a_h^\tau)^\top \Lambda_h^{-1}\phi(x, a) \\
&= \|\phi(x, a)\|_{\Lambda_h^{-1}}^2 - \lambda\phi(x, a)^\top (\Lambda_h^{-1})^2 \phi(x, a).
\end{aligned}
$$

Then, if we denote the event

$$
\mathcal{E}_h(x, a) := \{\sum_{\tau \in \mathcal{D}} \left\langle \phi(x, a), \Lambda_h^{-1}\phi_h^\tau \right\rangle \xi_h^\tau(V_{h+1}^*) > H\sqrt{\log\left(\frac{2H^2M}{\delta}\right)} \|\phi(x, a)\|_{\Lambda_h^{-1}}\},
$$

then it follows that

$$\mathbb{P}(\mathcal{E}_h(x,a)) = \int \mathbb{P}\left(\mathcal{E}_h(x,a)|\mathcal{D}_h = D_h\right) d\mu(D_h) \le \frac{\delta}{H^2 M},$$

where the last inequality holds for any fixed $D_h$. By a union bound over all $\phi(x,a)$, we know that with probability at least $1 - \delta/H^2$, for all $(x,a) \in \mathcal{S} \times \mathcal{A}$, we have

$$\sum_{\tau \in \mathcal{D}} \left\langle \phi(x,a), \Lambda_h^{-1} \phi_h^\tau \right\rangle \xi_h^\tau(V_{h+1}^*) \le H\sqrt{\log\left(\frac{2H^2 M}{\delta}\right)} \|\phi(x,a)\|_{\Lambda_h^{-1}}.$$

By similar induction procedure, we can set $\Gamma_h(\cdot,\cdot) = O\left(H\sqrt{\log\left(\frac{2H^2 M}{\delta}\right)}\right) \|\phi(x,a)\|_{\Lambda_h^{-1}}$, where we require $K \ge \tilde{\Omega}\left(d^2 H^2/\kappa\right)$ and $\lambda = 1/d$ to make the advantage function and (a) non-dominating. Then, the theorem follows from Lemma 2. $\qquad\square$

## E   PROOF OF LINPEVI-ADV+

In this section, we present the proof of Theorem 2.

### E.1   ANALYSIS OF THE VARIANCE ESTIMATION ERROR

First, we analyze the estimation error of the conditional variance estimator. We recall that we estimate $[\mathbb{V}_h V_{h+1}^*](x,a)$ based on dataset $\mathcal{D}'$ as

$$\widehat{\sigma}_h^2(x,a) = \max\left\{1, \underbrace{\left[\left\langle \phi^\top(x,a), \widetilde{\beta}_{h,2}\right\rangle\right]_{[0,H^2]} - \left[\left\langle \phi^\top(x,a), \widetilde{\beta}_{h,1}\right\rangle\right]_{[0,H]}^2}_{\mathbb{B}_h(x,a)} - \tilde{O}\left(\frac{dH^3}{\sqrt{K\kappa}}\right)\right\}.$$

**Lemma 5** (Variance Estimator). *Under Assumption 1, if $K \ge \tilde{\Omega}(d^2 H^2/\kappa)$, then with probability at least $1 - \delta$, for all $(x,a,h) \in \mathcal{S} \times \mathcal{A} \times [H]$, we have*

$$[\mathbb{V}_h V_{h+1}^*](x,a) - \tilde{O}\left(\frac{dH^3}{\sqrt{K\kappa}}\right) \le \widehat{\sigma}_h^2(x,a) \le [\mathbb{V}_h V_{h+1}^*](x,a) \qquad (21)$$

*Proof.* We first bound the difference between $\mathbb{B}_h(x,a)$ and $[\mathrm{Var}_h \widehat{V}_{h+1}'](x,a)$:

$$\left|\left\langle \phi^\top(x_h,a_h), \widetilde{\beta}_{h,2}\right\rangle_{[0,H^2]} - \left[\left\langle \phi^\top(x_h,a_h), \widetilde{\beta}_{h,1}\right\rangle\right]_{[0,H]}^2 - [\mathbb{P}_h(\widehat{V}_{h+1}')^2](x_h,a_h) - ([\mathbb{P}_h\widehat{V}_{h+1}'(x_h,a_h)])^2\right|$$

$$\le \left|\left\langle \phi^\top(x_h,a_h), \widetilde{\beta}_{h,2}\right\rangle_{[0,H^2]} - [\mathbb{P}_h(\widehat{V}_{h+1}')^2](x_h,a_h)\right| + \left[\left\langle \phi^\top(x_h,a_h), \widetilde{\beta}_{h,1}\right\rangle\right]_{[0,H]}^2 - ([\mathbb{P}_h\widehat{V}_{h+1}'(x_h,a_h)])^2$$

$$\le \underbrace{\left|\left\langle \phi^\top(x_h,a_h), \widetilde{\beta}_{h,2}\right\rangle - [\mathbb{P}_h(\widehat{V}_{h+1}')^2](x_h,a_h)\right|}_{(a)} + 2H\underbrace{\left|\left\langle \phi^\top(x_h,a_h), \widetilde{\beta}_{h,1}\right\rangle - [\mathbb{P}_h\widehat{V}_{h+1}'(x_h,a_h)]\right|}_{(b)}.$$

We note that both (a) and (b) are analysis of the regular value-target ridge regression, with target $(\widehat{V}_{h+1}')^2$ and $(\widehat{V}_{h+1}')$, respectively. The analysis thus follows the same line of those presented in Appendix D when we deal with the correlated advantage part, except that we invoke Lemma 9 with range $H$ for $\widehat{V}_{h+1}'$ and $H^2$ for $(\widehat{V}_{h+1}')^2$. We omit the details here for simplicity and present the results directly: with probability at least $1 - \delta/2$, for all $(x,a,h) \in \mathcal{S} \times \mathcal{A} \times [H]$,

$$\left|\mathbb{B}_h(x,a) - [\mathrm{Var}_h \widehat{V}_{h+1}'](x,a)\right| \le (a) + 2H(b) \le \tilde{O}\left(\frac{dH^2}{\sqrt{K\kappa}}\right). \qquad (22)$$

We then use Theorem 2 and Lemma 13 to show that with probability at least $1 - \delta/2$, for all $(x, a, h) \in \mathcal{S} \times \mathcal{A} \times [H]$,

$$
\left\| \mathrm{Var}_h \widehat{V}'_{h+1} - \mathrm{Var}_h V^*_{h+1} \right\| (x, a)
$$

$$
\leq \left\| \mathbb{P}_h \left( (\widehat{V}'_{h+1})^2 - (V^*_{h+1})^2 \right) \right\| (x, a) + \left\| \left( \mathbb{P}_h \widehat{V}'_{h+1} \right)^2 - \left( \mathbb{P}_h V^*_{h+1} \right)^2 \right\| (x, a) \tag{23}
$$

$$
\leq 2H \left\| \widehat{V}'_{h+1} - V^*_{h+1} \right\| (x, a) + 2H \left\| \mathbb{P}_h \widehat{V}'_{h+1} - \mathbb{P}_h V^*_{h+1} \right\| (x, a) \leq \tilde{O} \left( \frac{\sqrt{d} H^3}{\sqrt{K\kappa}} \right).
$$

With a union bound over the estimations in Eqn. (22) and Eqn. (23), we know that with probability at least $1 - \delta$, the following derivations hold. First, by triangle inequality, we have

$$
\left\| \mathbb{B}_h - \mathrm{Var}_h V^*_{h+1} \right\| (x, a) \leq \left\| \mathbb{B}_h - \mathrm{Var}_h \widehat{V}'_{h+1} \right\| (x, a) + \left\| \mathrm{Var}_h \widehat{V}'_{h+1} - \mathrm{Var}_h V^*_{h+1} \right\| (x, a)
$$

$$
\leq \tilde{O} \left( \frac{d H^3}{\sqrt{K\kappa}} \right),
$$

where we use Eqn. (22) and Eqn. (23) in the last inequality. This shows that $\mathbb{B}_h(x, a) - \tilde{O} \left( \frac{d H^3}{\sqrt{K\kappa}} \right) \leq [\mathrm{Var}_h V^*_{h+1}](x, a)$ and the second inequality of the lemma follows from the fact that $\max\{1, \cdot\}$ preserves the order of two numbers. On the other hand, we note that $\max\{1, \cdot\}$ is non-expansive, meaning that $|\max\{1, a\} - \max\{1, b\}| \leq |a - b|$. Then, we have

$$
\left\| \widehat{\sigma}^2_h - \mathbb{V}_h V^*_{h+1} \right\| (x, a) \leq \left\| \widehat{\sigma}^2_h - \mathbb{V}_h \widehat{V}'_{h+1} \right\| (x, a) + \left\| [\mathbb{V}_h \widehat{V}'_{h+1}] - [\mathbb{V}_h V^*_{h+1}] \right\| (x, a)
$$

$$
\leq \left\| \mathbb{B}_h - \tilde{O} \left( \frac{d H^3}{\sqrt{K\kappa}} \right) - \mathrm{Var}_h \widehat{V}'_{h+1} \right\| (x, a) + \left\| \mathrm{Var}_h \widehat{V}'_{h+1} - \mathrm{Var}_h V^*_{h+1} \right\| (x, a)
$$

$$
\leq \tilde{O} \left( \frac{d H^2}{\sqrt{K\kappa}} \right) + \tilde{O} \left( \frac{d H^3}{\sqrt{K\kappa}} \right) + \tilde{O} \left( \frac{d H^3}{\sqrt{K\kappa}} \right)
$$

$$
= \tilde{O} \left( \frac{d H^3}{\sqrt{K\kappa}} \right),
$$

where the third inequality follows from Eqn. (22) and Eqn. (23). $\qquad \square$

### E.2 PROOF OF THEOREM 2

The proof idea is similar to that of LinPEVI-ADV, except that we need to additionally estimate the conditional variance and apply the Bernstein-type Lemma 10 for the reference function. To simplify the presentation, we will focus on the uncertainty of the reference function as it is dominating, and focus on determining the threshold. To this end, we will omit the constants and logarithmic terms by $\tilde{O}(\cdot)$ throughout the proof.

*Proof of Theorem 2.* We still start with the following decomposition: for a function $\|g_{h+1}\|_\infty \leq R - 1$, we invoke Lemma 3:

$$
\left| \mathcal{T}_h g_{h+1} - \widehat{\mathcal{T}}_h g_{h+1} \right| (x, a)
$$

$$
\leq \underbrace{\sqrt{\lambda} d R \|\phi(x, a)\|_{\Sigma_h^{-1}}}_{(a)} + \underbrace{\left\| \sum_{\tau \in \mathcal{D}} \frac{\phi(x^\tau_h, a^\tau_h)}{\sqrt{[\bar{\mathbb{V}}_h \widehat{V}'_{h+1}](x^\tau_h, a^\tau_h)}} \cdot \xi^\tau_h(g_{h+1}) \right\|_{\Sigma_h^{-1}}}_{(b)} \|\phi(x, a)\|_{\Sigma_h^{-1}}.
$$

Similarly, we can set $\lambda = 1/H^2$ to ensure that $(a) \leq \beta_2 \|\phi(x, a)\|_{\Sigma_h^{-1}} = \tilde{O} \left( \sqrt{d} \right) \|\phi(x, a)\|_{\Sigma_h^{-1}}$, so we focus on the analysis of (b). For the reference function, as $[\bar{\mathbb{V}}_h \widehat{V}'_{h+1}](x^\tau_h, a^\tau_h) \geq 1$, we know that $\xi^\tau_h(V^*_{h+1}) \leq H$. Then we consider the filtration $\mathcal{F}_{\tau-1, h} = \sigma \left( \{(x^j_h, a^j_h)\}^\tau_{j=1, j \in \mathcal{D}} \cup \{(r^j_h, x^j_{h+1})\}^{\tau-1}_{j=1, j \in \mathcal{D}} \right)$ where $\sigma(\cdot)$ denotes the $\sigma$-algebra generated by the

random variables. As $[\bar{\mathbb{V}}_h \widehat{V}'_{h+1}](x_h^\tau, a_h^\tau)$ is independent of $\mathcal{D}$, $\xi_h^\tau(V_{h+1}^*)$ is mean-zero conditioned on $\mathcal{F}_{\tau-1,h}$. We proceed to estimate the conditional variance:

$$
\begin{aligned}
\mathbb{D}_{\tau-1,h}[\xi_h^\tau(V_{h+1}^*)] &= \frac{\mathbb{D}_{\tau-1,h}[V_{h+1}^*(x_{h+1}^\tau)]}{[\bar{\mathbb{V}}_h \widehat{V}'_{h+1}](x_h^\tau, a_h^\tau)} \leq \frac{[\mathbb{V}_h V_{h+1}^*(x_{h+1}^\tau)]}{[\bar{\mathbb{V}}_h \widehat{V}'_{h+1}](x_h^\tau, a_h^\tau)} \\
&\leq 1 + \frac{\tilde{O}\left(\frac{dH^3}{\sqrt{K\kappa}}\right)}{[\bar{\mathbb{V}}_h \widehat{V}'_{h+1}](x_h^\tau, a_h^\tau) - \tilde{O}\left(\frac{dH^3}{\sqrt{K\kappa}}\right)} \\
&\leq 1 + 2\tilde{O}\left(\frac{dH^3}{\sqrt{K\kappa}}\right) = O(1),
\end{aligned}
$$

where in the first equality, we use the fact that $[\bar{\mathbb{V}}_h \widehat{V}'_{h+1}](\cdot, \cdot)$ is independent of $\mathcal{D}$, and in the last inequality, we use $K \geq \tilde{\Omega}\left(d^2 H^6/\kappa\right)$ to ensure that $[\bar{\mathbb{V}}_h \widehat{V}'_{h+1}](x_h^\tau, a_h^\tau) - \tilde{O}\left(\frac{dH^3}{\sqrt{K\kappa}}\right) \geq \frac{1}{2}$. Then, we can directly invoke Lemma 10 to obtain that:

$$
\left\| \sum_{\tau \in \mathcal{D}} \phi(x_h^\tau, a_h^\tau) \cdot \xi_h^\tau(V_{h+1}^*) \right\|_{\Sigma_h^{-1}} \leq \tilde{O}\left(\sqrt{d}\right).
$$

Similar to the proof of LinPEVI-ADV, the uncertainty of the advantage function is non-dominating for sufficiently large $K$ (determined later) so we can set

$$
\Gamma_h(\cdot, \cdot) = \tilde{O}\left(\sqrt{d}\right) \|\phi(x, a)\|_{\Sigma_h^{-1}}.
$$

Moreover, by Lemma 5, we have $[\bar{\mathbb{V}}_h \widehat{V}'_{h+1}](x_h^\tau, a_h^\tau) \leq [\mathbb{V}_h V_{h+1}^*](x_h^\tau, a_h^\tau) \leq H^2$, which implies that

$$
\left( \sum_{\tau \in \mathcal{D}} \frac{\phi_h^\tau (\phi_h^\tau)^\top}{[\bar{\mathbb{V}}_h \widehat{V}'_{h+1}](x_h^\tau, a_h^\tau)} + \lambda I_d \right)^{-1} \preceq \left( \sum_{\tau \in \mathcal{D}} \frac{\phi_h^\tau (\phi_h^\tau)^\top}{[\mathbb{V}_h V_{h+1}^*](x_h^\tau, a_h^\tau)} + \lambda I_d \right)^{-1} \preceq H^2 \left( \sum_{\tau \in \mathcal{D}} \phi_h^\tau (\phi_h^\tau)^\top + \lambda I_d \right)^{-1}.
$$

This implies that

$$
\|\phi(x, a)\|_{\Sigma_h^{-1}} \leq \|\phi(x, a)\|_{\Sigma_h^{*-1}} \leq H \|\phi(x, a)\|_{\Lambda_h^{-1}}, \qquad \forall (x, a).
$$

Following the same induction analysis procedure of the proof of Theorem 1, we know that $\left\| \widehat{V}_{h+1} - V_{h+1}^* \right\|_\infty \leq \tilde{O}\left(\frac{\sqrt{d}H^2}{\sqrt{K\kappa}}\right)$. Using the standard $\epsilon$-covering argument and Lemma 9, we know that we can set

$$
b_{1,h}(x, a) = \tilde{O}\left(\frac{d^{3/2}H^2}{\sqrt{K\kappa}}\right) \|\phi(x, a)\|_{\Sigma_h^{-1}}.
$$

To make it non-dominating, we require that $K \geq \tilde{\Omega}\left(d^2 H^4/\kappa\right)$. Moreover, to make (a) $= \sqrt{\lambda d} R \|\phi(x, a)\|_{\Sigma_h^{-1}}$ non-dominating, we set $\lambda = 1/H^2$. Then, the theorem follows from Lemma 2. $\qquad \square$

## F   PROOF OF MARKOV GAME

*Proof of Theorem 5.* The techniques developed for MDPs are readily extended to the MGs by decoupling the estimations into the max-player part and min-player part so we start with the following lemma, which is a counterpart of Lemma 2.

**Lemma 6** (Decomposition Lemma for MG). *Under the condition that the functions $\Gamma_h : \mathcal{S} \times \mathcal{A} \times \mathcal{B} \to \mathbb{R}$ in Algorithms 3 and 4 satisfying*

$$
\begin{aligned}
|\mathcal{T}_h \underline{V}_{h+1}(x, a, b) - \widehat{\mathcal{T}}_h \underline{V}_{h+1}(x, a, b)| &\leq \underline{\Gamma}_h(x, a, b), \\
|\mathcal{T}_h \overline{V}_{h+1}(x, a, b) - \widehat{\mathcal{T}}_h \overline{V}_{h+1}(x, a, b)| &\leq \overline{\Gamma}_h(x, a, b),
\end{aligned}
$$

*for any* $(x, a, b, h) \in \mathcal{S} \times \mathcal{A} \times \mathcal{B} \times [H]$, *then for Algorithms 4 and 3, we have*

$$V_1^{*,\widehat{\nu}}(x) - V_1^*(x) \le \overline{V}_1(x) - V_1^*(x) \le 2 \sup_\nu \sum_{h=1}^H \mathbb{E}_{\pi^*,\nu}[\overline{\Gamma}_h(x, a, b) \mid x_1 = x],$$

$$V_1^*(x) - V_1^{\widehat{\pi},*}(x) \le V_1^*(x) - \underline{V}_1(x) \le 2 \sup_\pi \sum_{h=1}^H \mathbb{E}_{\pi,\nu^*}[\underline{\Gamma}_h(x, a, b) \mid x_1 = x].$$

*Furthermore, we can obtain that for any* $x \in \mathcal{S}$,

$$V_1^{*,\widehat{\nu}}(x) - V_1^{\widehat{\pi},*}(x) = V_1^{*,\widehat{\nu}}(x) - V_1^*(x) + V_1^*(x) - V_1^{\widehat{\pi},*}(x)$$

$$\le 2 \sup_\nu \sum_{h=1}^H \mathbb{E}_{\pi^*,\nu}[\overline{\Gamma}_h(x, a, b) \mid x_1 = x] + 2 \sup_\pi \sum_{h=1}^H \mathbb{E}_{\pi,\nu^*}[\underline{\Gamma}_h(x, a, b) \mid x_1 = x].$$

*Proof.* See Appendix A in Zhong et al. (2022) for a detailed proof. □

Therefore, it suffices to determine the $\Gamma_h(\cdot, \cdot, \cdot)$ that establishes pessimism. Before continuing, we first prove Theorem 7, which is required in our subsequent analysis.

*Proof of Theorem 7.* We first note that Lemma 3 is constructed for (weighted) ridge regression and can be applied to linear MG by replacing $\phi(x, a) \in \mathbb{R}^d$ with $\phi(x, a, b)$ accordingly. Therefore, for a function $g : \mathcal{S} \times \mathcal{A} \times \mathcal{B} \to [0, H-1]$, we have

$$\left| \mathcal{T}_h g_{h+1} - \overline{\mathcal{T}}_h g_{h+1} \right| (x, a, b) \lesssim \left\| \sum_{\tau \in \mathcal{D}} \frac{\phi(x_h^\tau, a_h^\tau, b_h^\tau)}{\overline{\sigma}_h(x_h^\tau, a_h^\tau, b_h^\tau)} \cdot \overline{\xi}_h^\tau(g_{h+1}) \right\|_{\overline{\Sigma}_h^{-1}} \|\phi(x, a)\|_{\overline{\Sigma}_h^{-1}}$$

$$\left| \mathcal{T}_h g_{h+1} - \underline{\mathcal{T}}_h g_{h+1} \right| (x, a, b) \lesssim \left\| \sum_{\tau \in \mathcal{D}} \frac{\phi(x_h^\tau, a_h^\tau, b_h^\tau)}{\underline{\sigma}_h(x_h^\tau, a_h^\tau, b_h^\tau)} \cdot \underline{\xi}_h^\tau(g_{h+1}) \right\|_{\underline{\Sigma}_h^{-1}} \|\phi(x, a)\|_{\underline{\Sigma}_h^{-1}}. \tag{24}$$

where $\overline{\Sigma}_h = \sum_{\tau \in \mathcal{D}} \frac{\phi(x_h^\tau, a_h^\tau, b_h^\tau)\phi(x_h^\tau, a_h^\tau, b_h^\tau)^\top}{\overline{\sigma}_h^2(x_h^\tau, a_h^\tau, b_h^\tau)} + \lambda I_d$, $\overline{\xi}_h^\tau(g_{h+1}) = \frac{r_h^\tau + g_{h+1}(x_{h+1}^\tau) - \mathcal{T}_h g_{h+1}(x_h^\tau, a_h^\tau, b_h^\tau)}{\overline{\sigma}_h(x_h^\tau, a_h^\tau, b_h^\tau)}$, and $\underline{\Sigma}_h, \underline{\xi}_h^\tau(g_{h+1})$ are defined similarly for $\underline{\sigma}_h(\cdot, \cdot, \cdot)$. Moreover, we again omit the $\sqrt{\lambda}dH$ as we can set $\lambda$ sufficiently small (determined later).

For LinPMVI-ADV, we set $\overline{\sigma}_h = \underline{\sigma}_h = 1$ so $\overline{\Sigma}_h = \underline{\Sigma}_h = \Lambda_h$. By the reference-advantage decomposition for MG (Eqn. (B.2)), it suffices to focus on the reference function with the Nash value $V_{h+1}^*$ where Lemma 9 can be applied directly:

$$\left| \left( \mathcal{T}_h \overline{V}_{h+1} - \overline{\mathcal{T}}_h \overline{V}_{h+1} \right) (x, a, b) \right| \lesssim \tilde{O}\left( \sqrt{d}H \right) \|\phi(x, a, b)\|_{\Lambda_h^{-1}} = \Gamma_h(x, a, b),$$

$$\left| \left( \mathcal{T}_h \underline{V}_{h+1} - \underline{\mathcal{T}}_h \underline{V}_{h+1} \right) (x, a, b) \right| \lesssim \tilde{O}\left( \sqrt{d}H \right) \|\phi(x, a, b)\|_{\Lambda_h^{-1}} = \Gamma_h(x, a, b).$$

We follow the same induction analysis procedure of Theorem 1 to obtain that $\left\| \overline{V}_{h+1} - V_{h+1}^* \right\|_\infty \le \tilde{O}\left( \frac{H(H-h)}{\sqrt{K\kappa}} \right)$ and $\left\| \underline{V}_{h+1} - V_{h+1}^* \right\|_\infty \le \tilde{O}\left( \frac{H(H-h)}{\sqrt{K\kappa}} \right)$. By standard $\epsilon$-covering argument and Lemma 9, we can set

$$b_{1,h}(x, a, b) = O\left( \frac{d^{3/2}H^2\iota}{\sqrt{K\kappa}} \right) \|\phi(x, a, b)\|_{\Lambda_h^{-1}}.$$

To make it non-dominating, we require that $K \ge \tilde{\Omega}(d^2 H^2/\kappa)$. Also, to make $\sqrt{\lambda}dH \le \tilde{O}\left( \sqrt{d}H \right)$, it suffices to set $\lambda = 1$. Now we invoke Lemma 6 with $\Gamma_h = \overline{\Gamma}_h = \underline{\Gamma}_h$ to obtain that for any $x \in \mathcal{S}$,

$$V_1^{*,\widehat{\nu}}(x) - V_1^{\widehat{\pi},*}(x)$$

$$\le 2 \sup_\nu \sum_{h=1}^H \mathbb{E}_{\pi^*,\nu}[\Gamma_h(x, a, b) \mid x_1 = x] + 2 \sup_\pi \sum_{h=1}^H \mathbb{E}_{\pi,\nu^*}[\Gamma_h(x, a, b) \mid x_1 = x]$$

$$\le \tilde{O}(\sqrt{d}H) \cdot \left( \max_\nu \sum_{h=1}^H \mathbb{E}_{\pi^*,\nu}\|\phi(x_h, a_h, b_h)\|_{\Lambda_h^{-1}} + \max_\pi \sum_{h=1}^H \mathbb{E}_{\pi,\nu^*}\|\phi(x_h, a_h, b_h)\|_{\Lambda_h^{-1}} \right).$$

$\square$

With Theorem 7 in hand, similar to Lemma 5, we have the following lemma to control the variance estimation error and we omit the proof for simplicity.

**Lemma 7** (Variance Estimation Error for MGs). *Under Assumption 1, in Algorithm 3, if $K \geq \tilde{\Omega}(d^2 H^2/\kappa)$, then with probability at least $1 - \delta$, for all $(x, a, b, h) \in \mathcal{S} \times \mathcal{A} \times \mathcal{B} \times [H]$, we have*

$$[\mathbb{V}_h V_{h+1}^*](x, a, b) - \tilde{O}\left(\frac{dH^3}{\sqrt{K\kappa}}\right) \leq \overline{\sigma}_h(x, a, b) \leq [\mathbb{V}_h V_{h+1}^*](x, a, b)$$

$$[\mathbb{V}_h V_{h+1}^*](x, a, b) - \tilde{O}\left(\frac{dH^3}{\sqrt{K\kappa}}\right) \leq \underline{\sigma}_h(x, a, b) \leq [\mathbb{V}_h V_{h+1}^*](x, a, b).$$

Similar to the proof of Theorem 2, if $K \geq \tilde{\Omega}\left(d^2 H^6/\kappa\right)$, the conditional variances of $\overline{\xi}_h^\tau(V_{h+1}^*)$ and $\underline{\xi}_h^\tau(V_{h+1}^*)$ are $O(1)$ so it suffices to set $\overline{\Gamma}_h(\cdot, \cdot, \cdot) = \tilde{O}\left(\sqrt{d}\right) \|\phi(\cdot, \cdot, \cdot)\|_{\overline{\Sigma}_h^{-1}}$ and $\underline{\Gamma}_h(\cdot, \cdot, \cdot) = \tilde{O}\left(\sqrt{d}\right) \|\phi(\cdot, \cdot, \cdot)\|_{\underline{\Sigma}_h^{-1}}$. Moreover, because $[\mathbb{V}_h V_{h+1}^*](x, a, b) \leq H^2$, we have

$$\overline{\Sigma}_h^{-1} \preceq \Sigma_h^{*-1} \preceq H^2 \Lambda_h^{-1}, \qquad \underline{\Sigma}_h^{-1} \preceq \Sigma_h^{*-1} \preceq H^2 \Lambda_h^{-1}.$$

We can similarly establish $\left\|\overline{V}_{h+1} - V_{h+1}^*\right\|_\infty \leq \tilde{O}\left(\frac{\sqrt{d}H^2}{\sqrt{K\kappa}}\right)$ and $\left\|\underline{V}_{h+1} - V_{h+1}^*\right\|_\infty \leq \tilde{O}\left(\frac{\sqrt{d}H^2}{\sqrt{K\kappa}}\right)$, respectively. It suffices to set $\overline{b}_{1,h}(\cdot, \cdot, \cdot) = \tilde{O}\left(d^{3/2} H^2/\sqrt{K\kappa}\right) \|\phi(\cdot, \cdot, \cdot)\|_{\overline{\Sigma}_h^{-1}}$ and $\underline{b}_{1,h}(\cdot, \cdot, \cdot) = \tilde{O}\left(d^{3/2} H^2/\sqrt{K\kappa}\right) \|\phi(\cdot, \cdot, \cdot)\|_{\underline{\Sigma}_h^{-1}}$ where $K \geq \tilde{\Omega}(d^2 H^4/\kappa)$ is sufficient to make them non-dominating. Moreover, we need to set $\lambda = 1/H^2$ to make $\sqrt{\lambda d H} \leq \tilde{O}\left(\sqrt{d}\right)$. The theorem then follows from Lemma 6. $\square$

## G  PROOF OF LOWER BOUNDS

We only provide the proof of the lower bound for MGs, and the lower bound for MDP follows from the similar argument (see Remark 1 for details). In particular, we remark that the $\sqrt{d}$-dependency does not contradict Theorem 4 as the feature size of the constructed instances is exponentially in $d$.

*Proof of Theorem 6.* Our proof largely follows Zanette et al. (2021); Yin et al. (2022). We construct a family of MGs $\mathcal{M} = \{\mathcal{M}_u\}_{u \in \mathcal{U}}$, where $\mathcal{U} = \{u = (u_1, \ldots, u_H) \mid u_h \in \{-1, 1\}^{d-2}, \forall h \in [H]\}$. For any fixed $u \in \mathcal{U}$, the associated MDP $\mathcal{M}_u$ is defined by

**State space.** The state space $\mathcal{S} = \{-1, +1\}$.

**Action space.** The action space $\mathcal{A} = \mathcal{B} = \{-1, 0, 1\}^{d-2}$.

**Feature map.** The feature map $\phi : \mathcal{S} \times \mathcal{A} \times \mathcal{B} \to \mathbb{R}^d$ defined by

$$\phi(1, a, b) = \begin{pmatrix} \frac{a}{\sqrt{2d}} \\ \frac{1}{\sqrt{2}} \\ 0 \end{pmatrix} \in \mathbb{R}^d, \quad \phi(-1, a, b) = \begin{pmatrix} \frac{a}{\sqrt{2d}} \\ 0 \\ \frac{1}{\sqrt{2}} \end{pmatrix} \in \mathbb{R}^d.$$

**Transition kernel.** Let

$$\mu_h(1) = \mu_h(-1) = \begin{pmatrix} \mathbf{0}_{d-2} \\ \frac{1}{\sqrt{2}} \\ \frac{1}{\sqrt{2}} \end{pmatrix} \in \mathbb{R}^d.$$

By the assumption that $\mathbb{P}_h(s' \mid s, a, b) = \langle \phi(s, a, b), \mu_h(s') \rangle$, we know the MDP reduces to a homogeneous Markov chain with transition matrix

$$\mathbf{P} = \begin{pmatrix} \frac{1}{2} & \frac{1}{2} \\ \frac{1}{2} & \frac{1}{2} \end{pmatrix} \in \mathbb{R}^{2 \times 2}.$$

**Reward observation.** Let

$$\theta_{u,h} = \begin{pmatrix} \zeta u_h \\ \frac{1}{\sqrt{3}} \\ -\frac{1}{\sqrt{3}} \end{pmatrix} \in \mathbb{R}^d,$$

where $\zeta \in [0, \frac{1}{\sqrt{3d}}]$. By the assumption that $r_h(s, a, b) = \langle \phi(s, a, b), \theta_h \rangle$, we have

$$r_{u,h}(s, a, b) = \frac{s}{\sqrt{6}} + \frac{\zeta}{\sqrt{2d}} \langle a, u_h \rangle$$

We further assume the reward observation follows a Gaussian distribution:

$$R_{u,h}(s, a, b) \sim \mathcal{N}\left(\frac{s}{\sqrt{6}} + \frac{\zeta}{\sqrt{2d}} \langle a, u_h \rangle, 1\right).$$

**Date collection process.** Let $\{e_1, e_2, \cdots, e_{d-2}\}$ be the canonical bases of $\mathbb{R}^{d-2}$ and $\mathbf{0}_{d-2} \in \mathbb{R}^{d-2}$ be the zero vector. The behavior policy $\mu : \mathcal{S} \to \Delta_{\mathcal{A} \times \mathcal{B}}$ is defined as

$$\mu(e_j, \mathbf{0}_{d-2} \mid s) = \frac{1}{d}, \quad \forall j \in [d-2], \quad \mu(\mathbf{0}_{d-2}, \mathbf{0}_{d-2} \mid s) = \frac{2}{d}.$$

By construction, a Nash equilibrium of $\mathcal{M}_u$ is $(\pi^*, \nu^*)$ satisfying $\pi_h^*(\cdot) = u_h$ and $\nu_h^*(\cdot) = u_h$. Since the reward and transition are irrelevant with the min-player's policy, we use the notation $u^\pi = (u_1^\pi, \ldots, u_H^\pi) = (\text{sign}(\mathbb{E}_\pi[a_1]), \ldots, \text{sign}(\mathbb{E}_\pi[a_H]))$. Moreover, for any vector $v$, we denote by its $i$-th element $v[i]$. By the proof of Lemma 9 in Zanette et al. (2021) we know

$$V_u^* - V_u^{\pi,\nu} \geq \frac{\zeta}{\sqrt{2d}} \sum_{h=1}^{H} \sum_{i=1}^{d} \mathbf{1}\{u_h^\pi[i] - u_h[i]\}$$

$$:= \frac{\zeta}{\sqrt{2d}} D_H(u^\pi, u). \tag{25}$$

By Assouad's method (cf. Lemma 2.12 in Tsybakov (2009)), we have

$$\sup_{u \in \mathcal{U}} \mathbb{E}_u[D_H(u^\pi, u)] \geq \frac{(d-2)H}{2} \min_{u,u' \mid D_H(u,u')=1} \inf_{\psi} [\mathbb{P}_u(\psi \neq u) + \mathbb{P}_{u'}(\psi \neq u')],$$

where $\psi$ is the test function mapping from observations to $\{u, u'\}$. Furthermore, by Theorem 2.12 in Tsybakov (2009), we have

$$\min_{u,u' \mid D_H(u,u')=1} \inf_{\psi} [\mathbb{P}_u(\psi \neq u) + \mathbb{P}_{u'}(\psi \neq u')] \geq 1 - \left(\frac{1}{2} \max_{u,u' \mid D_H(u,u')=1} \text{KL}(\mathcal{Q}_u \| \mathcal{Q}_{u'})\right)^{1/2}, \tag{26}$$

where $\mathcal{Q}_u$ takes form

$$\mathcal{Q}_u = \prod_{k=1}^{K} \xi_1\left(s_1^k\right) \prod_{h=1}^{H} \mu\left(a_h^k, b_h^k \mid s_h^k\right) \left[R_{u,h}\left(s_h^k, a_h^k, b_h^k\right)\right]\left(r_h^k\right) \mathbb{P}_h\left(s_{h+1}^k \mid s_h^k, a_h^k, b_h^k\right),$$

where $\xi = [\frac{1}{2}, \frac{1}{2}]$ is the initial distribution.

$$\text{KL}(\mathcal{Q}_u \| \mathcal{Q}_{u'}) = K \cdot \sum_{h=1}^{H} \mathbb{E}_u[\log([R_{u,h}(s_h^1, a_h^1, b_h^1)](r_h^1) / [R_{u',h}(s_h^1, a_h^1, b_h^1)](r_h^1))]$$

$$= \frac{K}{d} \sum_{j=1}^{d-2} \text{KL}\left(\mathcal{N}\left(\frac{\zeta}{\sqrt{2d}} \langle e_j, u_h \rangle, 1\right) \middle\| \mathcal{N}\left(\frac{\zeta}{\sqrt{2d}} \langle e_j, u_h' \rangle, 1\right)\right)$$

$$= \frac{K}{d} \cdot \text{KL}\left(\mathcal{N}\left(\frac{\zeta}{\sqrt{2d}}, 1\right) \middle\| \mathcal{N}\left(\frac{-\zeta}{\sqrt{2d}}, 1\right)\right) = \frac{2K\zeta^2}{d^2}, \tag{27}$$

where the third equality uses the fact that $D_H(u, u') = 1$. Choosing $\zeta = \Theta(d/\sqrt{K})$, (25), (26), and (27) imply that

$$\inf_{\pi,\nu} \max_{u \in \mathcal{U}} \mathbb{E}_u[V_u^* - V_u^{\pi,\nu}] \gtrsim \frac{d\sqrt{d}H}{\sqrt{K}}. \tag{28}$$

Here $K \geq \Omega(d^3)$ ensures that $\zeta \leq \frac{1}{\sqrt{3d}}$. On the other hand, we have

$$\mathrm{Var}[V_{h+1}^*](s, a, b)$$

$$= \frac{1}{2}\left[V_{h+1}^*(-1) - \frac{1}{2}\left(V_{h+1}^*(+1) + V_{h+1}^*(-1)\right)\right]^2 + \frac{1}{2}\left[V_{h+1}^*(+1) - \frac{1}{2}\left(V_{h+1}^*(+1) + V_{h+1}^*(-1)\right)\right]^2$$

$$= \left[\frac{1}{2}\left(r_{h+1}^*(+1, u_{h+1}, u_{h+1}) - r_{h+1}^*(-1, u_{h+1}, u_{h+1})\right)\right]^2$$

$$= \frac{1}{6},$$

where the second equality follows from Bellman equation, and the last inequality uses the facts that $r_{u,h+1}(1, a, b) - r_{u,h+1}(-1, a, b) = \frac{2}{\sqrt{6}}$. By calculation, we have

$$\mathbb{E}\Sigma_h^* = \frac{3K}{2}\begin{pmatrix} \frac{2}{d^2}\mathbf{I}_{d-2} & \frac{1}{d\sqrt{d}}\mathbf{1}_{(d-2)\times 2} \\ \frac{1}{d\sqrt{d}}\mathbf{1}_{2\times(d-2)} & \mathbf{I}_2 \end{pmatrix} \in \mathbb{R}^{d \times d}.$$

By Gaussian elimination, we know $\|(\mathbb{E}\Sigma_h^*/K)^{-1}\| \leq d^2$. For all $h \in [H]$, $(s, a, b)$ and $K \geq \Omega(d^4 \log(2Hd/\delta))$, it holds with probability $1 - \delta$ that

$$\|\phi(s, a, b)\|_{(\Sigma_h^*)^{-1}}^2$$

$$\leq 2 \cdot \|\phi(s, a, b)\|_{(\mathbb{E}\Sigma_h^*)^{-1}}^2$$

$$= \frac{4}{3K}\left\{\frac{d}{4}a^\top\left\{\mathbf{I}_{d-2} + \frac{1}{d-2}\mathbf{1}_{(d-2)\times(d-2)}\right\}a - \frac{d}{2(d-2)}\mathbf{1}_{d-2}^\top a + \frac{d-1}{2(d-2)}\right\}$$

$$= \frac{4}{3K}\left\{\frac{d}{4} \cdot a^\top a + \frac{d}{4(d-2)}\left(1 - \mathbf{1}_{d-2}^\top a\right)^2 + \frac{1}{4}\right\}$$

$$\leq \frac{4}{3K}\left\{\frac{d(d-2)}{4} + \frac{d}{4(d-2)}\left(1 - \mathbf{1}_{d-2}^\top a\right)^2 + \frac{1}{4}\right\}$$

$$\leq \frac{4}{3K}\left\{\frac{d(d-2)}{4} + \frac{d(d-1)^2}{4(d-2)} + \frac{1}{4}\right\} \lesssim \frac{d^2}{K},$$

where the first inequality uses Lemma 13. Hence, we have

$$\max_{\nu}\sum_{h=1}^{H}\mathbb{E}_{\pi^*,\nu}\|\phi(s_h, a_h, b_h)\|_{(\Sigma_h^*)^{-1}} + \max_{\pi}\sum_{h=1}^{H}\mathbb{E}_{\pi,\nu^*}\|\phi(s_h, a_h, b_h)\|_{(\Sigma_h^*)^{-1}} \lesssim \frac{dH}{\sqrt{K}}. \tag{29}$$

for any $u \in \mathcal{U}$. Combining (28), (29), and the fact that $|V_u^* - V_u^{\pi,\nu}| \leq |V_u^{*,\nu} - V_u^{\pi,*}|$ for any $u \in \mathcal{U}$, we have with probability at least $1 - \delta$ that

$$\inf_{\pi,\nu}\max_{u \in \mathcal{U}}\mathbb{E}_u[V_u^{*,\nu} - V_u^{\pi,*}]$$

$$\geq c\sqrt{d} \cdot \Big(\max_{\nu}\sum_{h=1}^{H}\mathbb{E}_{\pi^*,\nu}\|\phi(s_h, a_h, b_h)\|_{(\Sigma_h^*)^{-1}} + \max_{\pi}\sum_{h=1}^{H}\mathbb{E}_{\pi,\nu^*}\|\phi(s_h, a_h, b_h)\|_{(\Sigma_h^*)^{-1}}\Big),$$

which concludes our proof. $\qquad\square$

**Remark 1.** *In our construction, the min-player will not affect the rewards and transitions. So by the similar derivations of (28) and (29), we can obtain $\inf_\pi \max_{u \in \mathcal{U}} \mathbb{E}_u[V_u^* - V_u^\pi] \gtrsim \frac{d\sqrt{d}H}{\sqrt{K}}$ and $\sum_{h=1}^{H}\mathbb{E}_{\pi^*}\|\phi(s_h, a_h)\|_{(\Sigma_h^*)^{-1}} \lesssim \frac{dH}{\sqrt{K}}$. Hence, we can establish the lower bound for MDP as desired.*

## H NUMERICAL SIMULATIONS

For completeness, we adopt a similar synthetic linear MDP instance that is also used in Min et al. (2021) and Yin et al. (2022), and redo the experiments to verify the theoretical findings. The adopted MDP has $\mathcal{S} = \{1, 2\}$, $\mathcal{A} = \{0, \cdots, 99\}$, and $d = 10$. For the feature, we apply binary encoding to represent $a \in \mathcal{A}$ by $\mathbf{a} \in \mathbb{R}^8$. For the last two bits of the feature, we define $\delta(x, a) = \mathbf{1}(x = 0, a = 0)$ where $\mathbf{1}$ is the indicator function. Then, the MDP is characterized as follows.

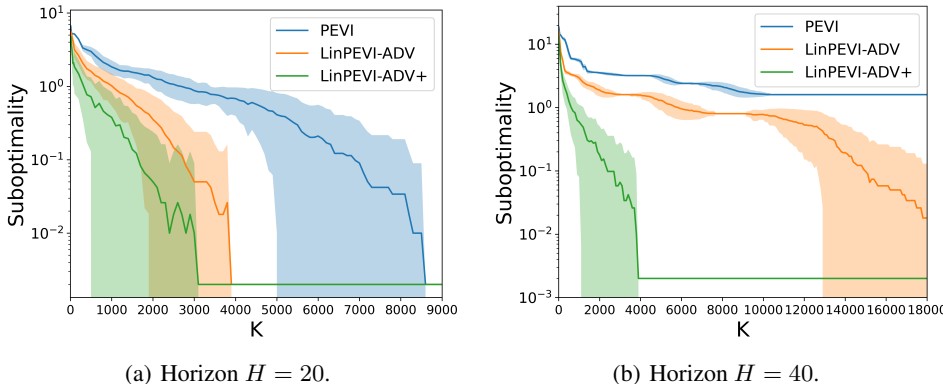

(a) Horizon $H = 20$.   (b) Horizon $H = 40$.

Figure 1: Suboptimality v.s. The number of trajectories $K$. The results are averaged aver 100 independent trails and the mean result is plotted as solid lines. The error bar area corresponds to the standard deviation.

- Feature mapping:
$$\phi(x, a) = (\mathbf{a}^\top, \delta(x, a), 1 - \delta(x, a))^\top \in \mathbb{R}^d.$$

- True measure $\nu_h$:
$$\nu_h(x) = (0, \cdots, 0, (1 - s) \oplus \alpha_h, x \oplus \alpha_h),$$

  where $\{\alpha_h\}_{h \in [H]}$ is a sequence of integers taking values in 0 or 1 generated randomly and fixed, and $\oplus$ is the XOR operator. The tranision is given by $\mathbb{P}_h(x'|x, a) = \langle \phi(x, a), \nu_h(x') \rangle$.

- Reward function: we define
$$\theta_h = (0, \cdots, 0, r, 1 - r) \in \mathbb{R}^{10}$$

  with $r = 0.9$ to obtain the mean reward $r_h(x, a) = \langle \phi(x, a), \theta_h \rangle$. Thr reward is then generated as a Bernoulli random variable.

- Behavior policy: always choose $a = 0$ with probability $p$, and other actions uniformly with $(1 - p)/99$. We choose $p = 0.5$.

- The initial state is chosen uniformly from $\mathcal{S}$.

- The regularization parameter $\lambda$ is set to be $0.1$ as suggested by Yin et al. (2022) and we estimate the value of the learned policy by 1000 i.i.d. trajectories where we also set the reward to be its mean during the this evaluation process.

Figure 1(a) and Figure 1(b) match our theoretical findings that a sharper bonus function leads to a smaller suboptimality. Therefore, LinPEVI-ADV+ achieves the best sample complexity, and PEVI performs worst. In particular, as $H$ increases, we can see that both LinPEVI-ADV and PEVI perform worse significantly, while the convergence rate of LinPEVI-ADV+ is rather stable. This demonstrates the power of variance information, which has been observed by the previous work on offline linear MDP (Min et al., 2021; Yin et al., 2022).

# I  PROOF OF AUXILIARY LEMMAS

*Proof of Lemma 3.* We add and subtract $\phi(x,a)^\top \Sigma_h^{-1}(\Sigma_h - \lambda I_d)w_h$ in the second equality to obtain that

$$
(\mathcal{T}_h f_{h+1})(x,a) - \left(\widehat{\mathcal{T}}_h f_{h+1}\right)(x,a) = \phi(x,a)^\top(w_h - \widehat{w}_h)
$$

$$
= \phi(x,a)^\top w_h - \phi(x,a)^\top \Sigma_h^{-1} \left( \sum_{\tau \in \mathcal{D}} \phi_h^\tau \cdot \frac{\left(r_h^\tau + f_{h+1}\left(x_{h+1}^\tau\right)\right)}{\widehat{\sigma}_h^2(x_h^\tau, a_h^\tau)} \right)
$$

$$
= \phi(x,a)^\top w_h - \phi(x,a)^\top \Sigma_h^{-1}(\Sigma_h - \lambda I_d)w_h
$$
$$
+ \phi(x,a)^\top \Sigma_h^{-1} \left( \sum_{\tau \in \mathcal{D}} \frac{\phi_h^\tau (\phi_h^\tau)^\top}{\widehat{\sigma}_h^2(x_h^\tau, a_h^\tau)} w_h \right) - \phi(x,a)^\top \Sigma_h^{-1} \left( \sum_{\tau \in \mathcal{D}} \phi_h^\tau \cdot \frac{\left(r_h^\tau + f_{h+1}\left(x_{h+1}^\tau\right)\right)}{\widehat{\sigma}_h^2(x_h^\tau, a_h^\tau)} \right)
$$

$$
= \lambda \phi(x,a)^\top \Sigma_h^{-1} w_h + \phi(x,a)^\top \Sigma_h^{-1} \left( \sum_{\tau \in \mathcal{D}} \phi_h^\tau \cdot \frac{\left(\mathcal{T}_h f_{h+1}(x_h^\tau, a_h^\tau) - r_h^\tau - f_{h+1}(x_{h+1}^\tau)\right)}{\widehat{\sigma}_h^2(x_h^\tau, a_h^\tau)} \right)
$$

$$
\leq \underbrace{\lambda \|w_h\|_{\Sigma_h^{-1}} \|\phi(x,a)\|_{\Sigma_h^{-1}}}_{\text{(i)}} + \underbrace{\left\| \sum_{\tau \in \mathcal{D}} \frac{\phi_h^\tau}{\widehat{\sigma}_h(x_h^\tau, a_h^\tau)} \cdot \xi_h^\tau(f_{h+1}) \right\|_{\Sigma_h^{-1}}}_{\text{(ii)}} \|\phi(x,a)\|_{\Sigma_h^{-1}}.
$$

This proves the first part of the lemma. Now suppose that $|f_{h+1}|$ is bounded by $H-1$. We have,

$$
\|w_h\| = \left\| \theta_h + \int_{\mathcal{S}} f_{h+1}(x')\mu_h(x')dx' \right\| \leq (1 + \max_{x'} f_{h+1}(x'))\sqrt{d} \leq H\sqrt{d},
$$

and

$$
\lambda \sqrt{w_h^\top \Sigma_h^{-1} w_h} \leq \|w_h\| \sqrt{\left\| \Sigma_h^{-1} \right\|} \leq \lambda \|w_h\| \sqrt{\lambda^{-1}} \leq \sqrt{\lambda d} H,
$$

where we use $\lambda_{\min}(\Sigma_h) \leq \lambda^{-1}$. Therefore, by setting $\lambda$ sufficiently small such that $\sqrt{\lambda d}H \leq \beta$, (i) is non-dominating and we can focus on bounding (ii). $\qquad \square$

# J  TECHNICAL LEMMAS

**Lemma 8** (Hoeffding's inequality (Wainwright, 2019)). *Let $X_1, \cdots, X_n$ be mean-zero independent random variables such that $|X_i| \leq \xi_i$ almost surely. Then, for any $t > 0$, we have*

$$
\mathbb{P} \left( \frac{1}{n} \sum_{i=1}^n X_i \geq t \right) \leq \exp \left( -\frac{2n^2 t^2}{\sum_{i=1}^n \xi_i^2} \right).
$$

**Lemma 9** (Hoeffding-type inequality for self-normalized process Abbasi-Yadkori et al. (2011)). *Let $\{\eta_t\}_{t=1}^\infty$ be a real-valued stochastic process and let $\{\mathcal{F}_t\}_{t=0}^\infty$ be a filtration such that $\eta_t$ is $\mathcal{F}_t$-measurable. Let $\{x_t\}_{t=1}^\infty$ be an $\mathbb{R}^d$-valued stochastic process where $x_t$ is $\mathcal{F}_{t-1}$ measurable and $\|x_t\| \leq L$. Let $\Lambda_t = \lambda I_d + \sum_{s=1}^t x_s x_s^\top$. Assume that conditioned on $\mathcal{F}_{t-1}$, $\eta_t$ is mean-zero and $R$-subGaussian. Then for any $\delta > 0$, with probability at least $1 - \delta$, for all $t > 0$, we have*

$$
\left\| \sum_{s=1}^t x_s \eta_s \right\|_{\Lambda_t^{-1}} \leq R\sqrt{d \log \left(1 + (tL)/\lambda\right) + 2\log(1/\delta)} \leq R\sqrt{d \log \left( \frac{\lambda + tL}{\lambda \delta} \right)} = \tilde{O}\left(R\sqrt{d}\right).
$$

**Lemma 10** (Bernstein-type inequality for self-normalized process Zhou et al. (2021)). *Let $\{\eta_t\}_{t=1}^\infty$ be a real-valued stochastic process and let $\{\mathcal{F}_t\}_{t=0}^\infty$ be a filtration such that $\eta_t$ is $\mathcal{F}_t$-measurable. Let $\{x_t\}_{t=1}^\infty$ be an $\mathbb{R}^d$-valued stochastic process where $x_t$ is $\mathcal{F}_{t-1}$ measurable and $\|x_t\| \leq L$. Let $\Lambda_t = \lambda I_d + \sum_{s=1}^t x_s x_s^\top$. Assume that*

$$
|\eta_t| \leq R, \mathbb{E}[\eta_t|\mathcal{F}_{t-1}] = 0, \mathbb{E}[\eta_t^2|\mathcal{F}_{t-1}] \leq \sigma^2.
$$

*Then for any $\delta > 0$, with probability at least $1 - \delta$, for all $t > 0$, we have*

$$
\left\| \sum_{s=1}^t \mathbf{x}_s \eta_s \right\|_{\Lambda_t^{-1}} \leq 8\sigma \sqrt{d \log \left(1 + \frac{tL^2}{\lambda d}\right) \cdot \log \left( \frac{4t^2}{\delta} \right)} + 4R \log \left( \frac{4t^2}{\delta} \right) = \tilde{O}\left( \sigma \sqrt{d} \right).
$$

**Lemma 11** ($\epsilon$-Covering Number (Jin et al., 2021c))**.** *For all $h \in [H]$ and all $\epsilon > 0$, let $\mathcal{N}(\cdot)$ be the covering number of the function space specified in (19), we have*

$$\log|\mathcal{N}(\epsilon; R, B, \lambda)| \leq d \cdot \log(1 + 4R/\epsilon) + d^2 \cdot \log(1 + 8d^{1/2}B^2/(\epsilon^2\lambda)).$$

**Lemma 12** (Lemma H.4 of Min et al. (2021))**.** *Let $\phi : \mathcal{S} \times \mathcal{A} \rightarrow \mathbb{R}^d$ satisfying $\|\phi(x, a)\| \leq C$ for all $(x, a) \in \mathcal{S} \times \mathcal{A}$. For any $K > 0$ and $\lambda > 0$, define $\bar{\mathbb{G}}_K = \sum_{k=1}^{K} \phi(x_k, a_k)\phi(x_k, a_k)^\top + \lambda I_d$ where $(x_k, a_k)$'s are i.i.d. samples from some distribution $\nu$ over $\mathcal{S} \times \mathcal{A}$. Let $\mathbb{G} = \mathbb{E}_v[\phi(x, a)\phi(x, a)^\top]$. Then, for any $\delta \in (0, 1)$, with probability at least $1 - \delta$, it holds that*

$$\left\| \frac{\bar{\mathbb{G}}_K^{-1}}{K} - \mathbb{E}_\nu[\frac{\bar{\mathbb{G}}_K^{-1}}{K}] \right\| \leq \frac{4\sqrt{2}C^2}{\sqrt{K}} \left( \log \frac{2d}{\delta} \right)^{1/2}.$$

**Lemma 13** (Lemma H.5 of Min et al. (2021))**.** *Let $\phi : \mathcal{S} \times \mathcal{A} \rightarrow \mathbb{R}^d$ satisfying $\|\phi(x, a)\| \leq C$ for all $(x, a) \in \mathcal{S} \times \mathcal{A}$. For any $K > 0$ and $\lambda > 0$, define $\bar{\mathbb{G}}_K = \sum_{k=1}^{K} \phi(x_k, a_k)\phi(x_k, a_k)^\top + \lambda I_d$ where $(x_k, a_k)$'s are i.i.d. samples from some distribution $\nu$ over $\mathcal{S} \times \mathcal{A}$. Let $\mathbb{G} = \mathbb{E}_v[\phi(x, a)\phi(x, a)^\top]$. Then, for any $\delta \in (0, 1)$, if $K$ satisfies that*

$$K \geq \max\left\{ 512C^4 \left\|\mathbb{G}^{-1}\right\|^2 \log\left( \frac{2d}{\delta} \right), 4\lambda \left\|\mathbb{G}^{-1}\right\| \right\}. \tag{30}$$

*Then with probability at least $1 - \delta$, it holds simultaneously for all $u \in \mathbb{R}^d$ that*

$$\|u\|_{\bar{\mathbb{G}}_K^{-1}} \leq \frac{2}{\sqrt{K}} \|u\|_{\mathbb{G}^{-1}}.$$

