# OpenReview forum: "Nearly Minimax Optimal Offline Reinforcement Learning with Linear Function Approximation: Single-Agent MDP and Markov Game"
_ICLR.cc/2023/Conference — ICLR 2023 poster_

### Official Review · Reviewer_BwgL · 2022-10-12

**Confidence:** 4
**Correctness:** 4
**Technical Novelty And Significance:** 3
**Empirical Novelty And Significance:** 3
**Recommendation:** 6

**Clarity, Quality, Novelty And Reproducibility:**

The paper is clear and involves some novelty techniques and constructions. I think the weakest point is writing and it requires decent effort to improve it before being accepted.

**Strength And Weaknesses:**

The result is sharp and improves upon the best-known results so far. However, I think the current writing hurt the quality of the paper a lot. Before recommending to accept, I think the following aspects need to be handled properly.
1. What's the point of Section 4? According to my understanding, it behaves worse than the algorithm introduced later in the next section, and also the algorithmic form looks very standard, if not totally the same as Jin et al. (2021c). Actually, I think the algorithmic idea here has already been covered in that paper. If I have misunderstood anything, please correct me by explicitly adding some discussion on this difference.
2. Regarding Section 5, it seems to me the main technical contribution is to remove the additional (implicit) assumption in Yin et al. (2022). Am I understanding this correctly? If so I think this is something the authors may want to emphasize earlier in the paper. Actually, it would be very helpful to add a table comparing the result in all the related work in offline RL with linear function approximation.
3. The authors may also want to highlight some of the technical contributions of this paper, and make a sharp distinction between the existing techniques and the ones you proposed (or improved). For example, I think the advantage decomposition has already been used by Jin et al. (2021c). As a classic technique, the authors may even want to put this into the preliminaries section and only keep the very original part in the remaining sections. This not only helps the readers to place the current paper in the whole literature but also makes it easier to evaluate the value of the paper.

**Summary Of The Paper:**

The paper proposed minimax near-optimal algorithms and matching lower bounds for linear function approximation in MDPs and MGs. In particular, it removes the independent assumption of sampling across different time steps as in Yin et al. (2022).

**Summary Of The Review:**

The paper proposed minimax near-optimal algorithms and matching lower bounds for linear function approximation in MDPs and MGs, and also removes some (explicit or implicit) assumptions in the literature. However, the writing needs to be improved before I recommend accepting.

---

> ### Author Response · Authors · 2022-11-15
> **Thanks for your review and constructive comments!**
>
> Thanks for your review and constructive comments!
>
> **Q1:** The point of Section 4.
>
> **In Section 4, we use a novel advantage-reference decomposition to tackle the temporal dependency and thus improve a $\sqrt{d}$ factor in the final regret bound.**
>
> We provide some detailed explanations as follows.
>
> - The bonus in Jin et al. (2021c) is $\mathcal{O}(dH \cdot \|\phi(\cdot, \cdot)\|_{\Lambda_h^{-1}} )$, which is suboptimal in terms of both $d$ and $H$, and we have discussed the technical challenge from the temporal dependency between different time steps in Section 3. Therefore, Section 4 is devoted to improving the $d$-dependency by designing an uncertainty decomposition technique in linear setting to address such a challenge.
> - We want to emphasize that (Jin et al., 2021c) **does NOT** use such a decomposition. To the best of our knowledge, this technique is completely new in the literature in offline linear RL.
> - In terms of the algorithmic form, you are correct! We only decompose the uncertainty in the **theoretical analysis**, so the new algorithm shares the SAME pseudo code with PEVI, except for **a smaller bonus**, scaling as $\mathcal{O}(\sqrt{d}H \cdot \|\phi(\cdot, \cdot)\|_{\Lambda_h^{-1}} )$, which leads to an improvement of $\sqrt{d}$ over the regret bound in (Jin et al., 2021c). In the revision, we only present the algorithmic form in the Appendix for completeness.
> - Indeed, this is also a difference between our decomposition technique and the counterpart in tabular setting. We also present a comparison at the end of Section 4. We believe that such an improvement from theoretical analysis leads to a relatively clean algorithmic form.
>
> **Q2:** The point of Section 5.
>
> **In Section 5, our focus is to leverage the variance information to further improve the dependency on $H$ by carefully handling the temporal dependency.**
>
> Following your constructive suggestions, we have revised the paper as follows.
> - We first illustrate the high-level ideas of the variance-weighted regression and then point out the limitation of existing approach when we take the temporal dependency into account, thus motivating our algorithmic designs;
> - We then present the main theoretical guarantee of this paper. In the revision, we have added two detailed comparisons in cyan, as well as a table (Table 1) comparing our results with the most related works.
>
> **Q3:** Technical Contribution
>
> Thanks for your suggestion! In the revision, we have modified the paper as follows to better illustrate our contributions.
> - At the end of Introduction, we clearly summarize the contributions of this work in red;
> - We also want to remark that to the best of our knowledge, the advantage decomposition is mainly confined to the tabular setting. Therefore, the technique developed in this paper is new in the literature of offline linear RL. In particular, we present a detailed comparison with the work on offline tabular RL at the end of Section 4;
> - Moreover, we present a detailed comparison with the related methods in the literature of offline linear MDP after presenting the main result of this paper in Section 5. We hope these clarify the main contributions of this work;
> - Finally, we reorganize the paper and use the linear MDP with finite features and the Markov game as examples to demonstrate the broad adaptability of the developed techniques.

---

> > ### Comment · Reviewer_BwgL · 2022-11-15
> > **Response**
> >
> > Thanks for the authors' response. I can now understand the technical contribution and adjust my score accordingly. I still suggest put the table earlier and compare with each of the baselines there in the introduction, at least briefly. This will help you place your contribution much better.

---

> > > ### Author Response · Authors · 2022-11-16
> > > **Thank you very much for looking through the revised version!**
> > >
> > > We highly appreciate your continued efforts in reviewing the revised paper and also the constructive suggestions! We will further revise the paper accordingly.

---

### Official Review · Reviewer_NJRa · 2022-10-24

**Confidence:** 2
**Correctness:** 3
**Technical Novelty And Significance:** 3
**Empirical Novelty And Significance:** Not applicable
**Recommendation:** 6

**Clarity, Quality, Novelty And Reproducibility:**

The paper is relatively well written but to me this paper is too dense for a 9 page papers.  It seems to me that the main contribution of the paper is design an algorithm that fixes the algorithm of (Yin et al. 2022). This seems a contribution that is good enough.

**Strength And Weaknesses:**

This paper proposes to correct an existing algorithm for offline MDP.  This seems a valid contribution if the result holds. I must admit, however, that I found the paper very hard to read, with lots of technical details and notations. This might be because I am not that familiar with the previous paper but I also think that it is because the paper wants to present too many results. Hence, it is not really self-contained and the results are barely commented or described.

For instance, the authors choose to add an extension to two-player zero-sum Markov games. I do not think that this brings new insight nor original ideas compared to the MDP case but that takes an extra 2 pages. As a result, there is no related work in the paper, and no real comment or explanation around the results.

Since this paper is mostly a theoretical paper that aims at correcting a bug in a previous analysis, I think that this analysis should be made clear. In the current version, I cannot be sure that the authors are also overlooking some steps of the proofs.


**Summary Of The Paper:**

This paper studies offline reinforcement learning problem for linear MDPs. In such a setting, the controller is given a trace of execution of the MDP with a fixed exploration policies. The controller uses this trace to compute a near-optimal policy. The objective of this paper is to provide an algorithm with provable regret guarantee. The authors study both the classical MDP setting and the two-player zero-sum Markov games.

For MDP, such an algorithm already exists in the literature but the authors of the current paper argue that their analysis do not hold. The authors therefore propose a new algorithm that fixes the bugs.  This algorithm is the adapted to the two player zero-sum Markov game.


**Summary Of The Review:**

The paper proposes an algorithm for offline RL. This algorithm is claimed to correct a bug from previous work. This result seems strong but the writing quality is insufficient.

---

> ### Author Response · Authors · 2022-11-15
> **Thanks for your review and constructive comments!**
>
> Thanks for your review and constructive comments!
>
> The focal point of this paper is to design an efficient algorithm without a restricted assumption on the independence between time steps as required by the existing work. Thank you for recognizing that the paper is overall well written and we agree that we had included too much content for a 9-page paper. We have revised the paper accordingly. In particular, according to your constructive suggestion, we have moved the Markov game part to the Appendix but only briefly illustrated the idea as an example of the extension of the developed techniques.
>
> Specifically, we would like to highlight the current structure of revision as follows.
>
> - Section 1: We introduce the background and motivate the problem;
>     - 1.0: We introduce the background of offlne RL, point out the limitation of existing work in the linear setting, thus motivating our work. Also, we present **a summary of contributions in red** at the end of this subsection;
>     - 1.1: We add some related works in this subsection. In particular, we point out that dealing with the statistical dependency between different time steps is one of the core problems in RL by **reviewing the efforts made in a long line of work in the literature.**
> - Section 2: We formally formulate the standard linear MDP and offline RL setup;
> - Section 3: We elaborate on the technical challenges from temporal dependency with more details.
>     - We show that the suboptimality bound of a pessimism-based algorithm is determined by the bonus function.
>     - The existing suboptimal bonus of PEVI comes from the uniform concentration required to deal with the temporal dependency;
>     - We then discuss the limitation of existing work for handling the temporal dependency and motivate our new technique to fix it.
> - Section 4: To address the issue in Section 3, we design an uncertainty decomposition technique, which is new in the literature of offline linear RL.
>     - We first show that the true Bellman operator and the estimated one are both affine in terms of the target function. This leads to an uncertainty decomposition provided in Eqn. (5);
>     - We illustrate the high-level ideas of the decomposition technique and conclude that it serves to avoid a $\sqrt{d}$-amplification of value function error;
>     - We present **a detailed comparison with the counterpart in tabular setting.**
> - Section 5: To further improve the dependence on horizon $H$, we adopt the variance-weighted regression in this section.
>     - We first illustrate the high-level idea of variance-weighted regression;
>     - We then point out the limitation of existing approaches when we do not ignore the temporal dependency, thus motivating our algorithmic design.
>     - We present the theoretical result of the proposed algorithm, followed by **a detailed comparison with existing works for interpretation**.
> - Section 6: We generalize the techniques developed for linear MDP to illustrate the broad adaptability of our methods.
>     - linear MDP with finite features;
>     - two-player zero-sum linear Markov games (with only the main ideas, and details deferred to the Apeendix).
> - Section 7: Conclusion.
>
> We also want to highlight the following efforts we have made to improve the readability of this paper.
>
> - Notation: we present a notation table in Appendix A for readers' reference;
> - We also reformat many contents of the paper to make it not too compact for the reader.

---

> ### Author Response · Authors · 2022-11-29
> **Follow up to Reviewer NJRa**
>
> Dear reviewer:
>
> Thank you for your insightful comments. Point-by-point responses have been provided and the corresponding updates have been made to the paper, which are marked by the highlighted text. We would sincerely appreciate it if the reviewer could let us know of any additional concerns, or consider improving the rating if we have addressed the concerns.
>
> Thanks

---

> > ### Comment · Reviewer_NJRa · 2022-11-30
> > **The new version leaves more space for discussion**
> >
> > The new version leaves more space for the discussion / related work. I think that this is a good and clarifies some of the contributions. The paper remains technical, which is probably unavoidable but is now clearer. I updated my score.
> >
> > A very minor detail: I do not think that the column "Additional Assumption" of Table 1 are very clear. Suggestion: use words instead (like "Needs additional assumption" for Yin et al. ? and "--" for the others.)

---

> > > ### Author Response · Authors · 2022-11-30
> > > **Thanks**
> > >
> > > Thanks for your re-evaluation and useful suggestion. We will further revise the paper accordingly.

---

### Official Review · Reviewer_iMxh · 2022-11-01

**Confidence:** 3
**Correctness:** 4
**Technical Novelty And Significance:** 3
**Empirical Novelty And Significance:** Not applicable
**Recommendation:** 6

**Clarity, Quality, Novelty And Reproducibility:**

Clearly written
Good quality
Novelty in the sample complexity results

**Strength And Weaknesses:**


1. The paper is very well written; the contribution is clear.
2. The algorithm is very similar to Yin et. al (2022), is the difference only in the analysis to consider temporal dependency?
3. While the proposed algorithms are efficient in terms of sample complexity, it is not clear to me how would one test the algorithm numerically. For example, the algorithm takes in the input $\beta_1 = \mathcal{O}(\sqrt{dH})$, is it just an artifact of the analysis and it can been any number say in [0,1].


**Summary Of The Paper:**

This paper presents algorithms for offline RL combined with linear function approximation. They consider both MDP and Markov games scenarios. They combine two techniques: reference-advantage decomposition and variance-reweighted ridge regression which have been previously used in offline RL literature.  The authors then establish matching lower bounds for the proposed algorithms. The proposed algorithms are computationally efficient while having the tighter theoretical guarantees.


**Summary Of The Review:**

This paper presents algorithms for offline MDP and MG using pessimism in a variance-reduction manner and variance-reweighted ridge regression. The presented bounds are tighter and matches the lower bounds. The techniques used in this paper have been used previously in offline RL.

---

> ### Author Response · Authors · 2022-11-15
> **Thanks for your review and positive evaluation!**
>
> Thanks for your review and positive evaluation!
>
> **Q1:** The algorithm is very similar to Yin et. al (2022), is the difference only in the analysis to consider temporal dependency?
>
> In terms of the result in offline linear MDP, the answer is yes. While Yin et. al (2022) achieves similar theoretical result by omitting the complicated dependency between different time steps, we aim to attain the statistical limit in the presence of such a challenging dependency with new ideas and analytic techniques. We would like to highlight that handling the statistical dependency between time steps is one of the core problems in the literature and we present a review of the efforts made by a long line of work in Secion 1.1 of the revision.
>
> Beyond the linear MDP considered in Yin et. al (2022), we also extend the techniques developed for linear MDP to linear MDPs with finite features and two-player zero-sum Markov games, thus demonstrating the broad adaptability of our techniques. We have stated our contribution more clearly at the end of Introduction (in red).
>
> **Q2:** While the proposed algorithms are efficient in terms of sample complexity, it is not clear to me how would one test the algorithm numerically. For example, the algorithm takes in the input $\beta_1 = \mathcal{O}(\sqrt{dH})$, is it just an artifact of the analysis and it can been any number say in [0,1].
>
> We agree with your opinion. Indeed, the theoretically sound algorithms (like PEVI, or algorithms in this paper) could be far too pessimistic for the encountered average instances, thus leading to a relatively poor performance. Since this work is mainly theory-oriented, the main purpose to adopt such a bonus function in the proposed algorithm **is to cover the corner cases, thus closing the gap to the information-theoretic lower bound**.
>
> When it comes to the real-world applications, one may treat $\beta$ as a hyper-parameter and tune it to the best performance for their customized needs.

---

### Decision · Program_Chairs · 2023-01-20

**Decision:**

Accept: poster

**Justification For Why Not Higher Score:**

- The paper's clarity still has room to improve
- Given the continued lack of clarity, we still are not 100% sure the paper is correct
- There are tons of papers of regret analysis of this or that problem --- it's not a completely new thing
- The paper is mostly of theoretical interest, e.g., there is no new killer application
- The technical gap did not seem especially hard to fill

**Justification For Why Not Lower Score:**

- Correcting a significant technical issue with a previous paper in an important area is important
- The analysis is non-trivial

**Metareview: Summary, Strengths And Weaknesses:**

This paper considers offline RL for linear MDPs. Here, the algorithm is given a trajectory from a control policy and the algorithm's goal is to compute another policy whose expected reward is as large as possible.  The paper provides an algorithm for accomplishing this task with a regret guarantee. It focuses on the standard MDP setting and presents an extension to two-player zero-sum Markov games.

This problem was previously studied in Yin et al. 2022, which claimed to provide a regret guarantee, but that paper made an inappropriate assumption about independence across time steps.  In the context of that previous paper, this paper fixes this issue and extends the results to the two-player zero-sum Markov game setting.

Strengths
- Having a correct proof of the claimed result from Yin et al. 2022 without its inappropriate temporal dependence assumption is important
- The extension is nice and more novel. It supports the idea that the techniques can be used elsewhere
- The analysis seems non-trivial and the bound it provides seems useful

Weaknesses
- The paper is not easy to follow. It has improved during the review process, but it remains sometimes difficult to understand where the paper is going and why
- A reasonably large part of the analysis appears taken from Yin et al. 2022. This seems necessary to enable the paper to be self-contained, but is still a downside.
- Given the complexity of the paper, the reviewing team was not able to check every detail of every proof. As such, we are not 100% sure that the new proof is bug-free.

**Note From Pc:**

if the above contains the word "oral" or "spotlight" please see: "oral" presentation means -> notable-top-5% and "spotlight" means -> notable-top-25%. As stated in our emails, we are disassociating presentation type from AC recommendations

**Summary Of Ac-Reviewer Meeting:**

During the pre-meeting rebuttal process, the reviewers read each others' reviews and engaged with the authors. This created a consensus. As such, during the meeting, we summarized our views of the paper. There was broad agreement on the points made in the meta-review above.